# SELF-DISTRIBUTION DISTILLATION: EFFICIENT UNCERTAINTY ESTIMATION

## ABSTRACT

Deep learning is increasingly being applied in safety-critical domains. For these scenarios it is important to know the level of uncertainty in a model's prediction to ensure that appropriate decisions are made by a system. Deep ensembles are the de-facto standard approach to obtaining various measures of uncertainty. However, ensembles normally significantly increase the resources required in both the training and deployment phases. Approaches have been developed that typically address the costs in one of these phases. In this work we propose a novel training approach, self-distribution distillation (S2D), which is able to efficiently, both in time and memory, train a single model that can estimate uncertainties in an integrated training phase. Furthermore it is possible to build ensembles of these models and apply ensemble distillation approaches, hierarchical distribution distillation, in cases where one is less limited by computational resources in the training phase, but still requires efficiency in the deployment phase. Experiments on CIFAR-100 showed that S2D models outperformed standard models and Monte-Carlo dropout. Additional out-of-distribution detection experiments on LSUN, Tiny ImageNet, SVHN showed that even a standard deep ensemble can be outperformed using S2D based ensembles and novel distilled models.

## 1 INTRODUCTION

Neural networks (NNs) have enjoyed much success in recent years achieving state-of-the-art performance on a large number of tasks within domains such as natural language processing (Vaswani et al., 2017), speech recognition (Hinton et al., 2012) and computer vision (Krizhevsky et al., 2012). Unfortunately, despite the prediction performance of NNs they are known to yield poor estimates of the uncertainties in their predictions—in *knowing what they do not know* (Lakshminarayanan et al., 2017; Guo et al., 2017). With the increasing application of neural network based systems in performing safety-critical tasks such as biometric identification (Schroff et al., 2015), medical diagnosis (De Fauw et al., 2018) or fully autonomous driving (Kendall et al., 2019), it becomes increasingly important to be able to robustly estimate the uncertainty in a model's prediction. By having access to accurate measures of prediction uncertainty, a system can act in a more safe and informed manner.

Ensemble methods, and related schemes, have become the standard approach for uncertainty estimation. Lakshminarayanan et al. (2017) proposed generating a deep (random-seed) ensemble by training each member model with a different initialisation and stochastic gradient descent (SGD). Not only does this ensemble perform significantly better than a standard trained NN, it also displays better predictive uncertainty estimates. Although simple to implement, training and deploying an ensemble results in a linear increase in the computational cost. Alternatively Gal & Ghahramani (2016) introduced the *Monte Carlo (dropout) ensemble* (MC ensemble) which at test time estimates predictive uncertainty by sampling members of an ensemble using dropout. Though this approach generally does not perform as well as a deep ensemble (given the same computational power and neglecting memory) (Lakshminarayanan et al., 2017), it is significantly cheaper to train as it integrates the ensemble generation method into training.

Despite ensemble generation methods being computationally more expensive, they have an important ability to decompose predictive (total) uncertainty into *data* and *knowledge uncertainty* (Depeweg et al., 2018; Gal & Ghahramani, 2016). Knowledge or *epistemic* uncertainty refers to the lack of knowledge or ignorance about the most optimal choice of model (parameters) (Hüllermeier

& Waegeman, 2021). As additional data is collected, the uncertainty in model parameters should decrease. This form of uncertainty becomes important whenever the model is tasked with making predictions for out-of-distribution data-points. For in-distribution inputs, it is expected that the trained model can return reliable predictions. On the other hand data or *aleatoric* uncertainty, represents inherent noise in the data being modelled, for example from overlapping classes. Even if more data is collected, this type of noise is inherent to the process and cannot be avoided or reduced (Malinin & Gales, 2018; Gal & Ghahramani, 2016; Ovadia et al., 2019) The ability to decompose and distinguish between these sources of uncertainty is important as it allows the cause of uncertainty in the prediction to be known and how it should be used in downstream tasks (Houlsby et al., 2011; Kirsch et al., 2019).

*Summary of contributions*: In this work we make two important contributions to NN classifier training and uncertainty prediction. First we introduce *self-distribution distillation* (S2D), a new general training approach that in an integrated, simultaneous fashion, trains a teacher ensemble and distribution distils the knowledge to a student. This integrated training allows the user to bypass training a separate expensive teacher ensemble while distribution distillation (Malinin et al., 2020) allows the student to capture the diversity and model a distribution over ensemble member predictions. Additionally, distribution distillation would give the student the ability to estimate both data and knowledge uncertainty in a single forward pass unlike standard NNs which inherently can not decompose predictive uncertainty, and unlike ensemble methods which can not perform the decomposition in a single pass. Second, we train an ensemble of these newly introduced models and investigate different distribution distillation techniques giving rise to *hierarchical distributions over predictions for uncertainty*. This approach is useful when there are no, or few, computational constraints in the training phase but still require robust uncertainties and efficiency in the deployment stage.

## 2 BACKGROUND AND RELATED WORK

This section describes two techniques for uncertainty estimation. First, ensemble methods for predictive uncertainty estimation will be viewed from a Bayesian viewpoint. Second, a specific form of distillation for efficient uncertainty estimation will be discussed.

### 2.1 ENSEMBLE METHODS

From a Bayesian perspective the parameters, $\boldsymbol{\theta}$, of a neural net are treated as random variables with some prior distribution $p(\boldsymbol{\theta})$. Together with the training data $\mathcal{D}$, this allows the posterior distribution $p(\boldsymbol{\theta}|\mathcal{D})$ to be derived. To obtain the predictive distribution over all classes $y \in \mathcal{Y}$ (for some input $\boldsymbol{x}^*$) marginalisation over $\boldsymbol{\theta}$ is required:

$$\mathrm{P}(y|\boldsymbol{x}^*, \mathcal{D}) = \mathbb{E}_{p(\boldsymbol{\theta}|\mathcal{D})}\Big[\mathrm{P}(y|\boldsymbol{x}^*, \boldsymbol{\theta})\Big] \tag{1}$$

Since finding the true posterior is intractable a variational approximation $p(\boldsymbol{\theta}|\mathcal{D}) \approx q(\boldsymbol{\theta})$ is made (Jordan et al., 1999; Blundell et al., 2015; Graves, 2011; Maddox et al., 2019). Furthermore, marginalising over all weight values remains intractable leading to a sampling ensemble, approximation method (Gal & Ghahramani, 2016; Lakshminarayanan et al., 2017):

$$\mathrm{P}(y|\boldsymbol{x}^*, \mathcal{D}) \approx \frac{1}{M} \sum_{m=1}^{M} \mathrm{P}(y|\boldsymbol{x}^*, \boldsymbol{\theta}^{(m)}), \ \boldsymbol{\theta}^{(m)} \sim q(\boldsymbol{\theta}) \tag{2}$$

Here, an ensemble generation method is required to obtain the predictive distribution and uncertainty. Two previously mentioned approaches to generate an ensemble are deep (naive) random-seed and MC-dropout ensemble[1]. Deep ensembles are based on training $M$ models on the same data but with different initialisations leading to functionally different solutions. On the other hand, a MC-dropout ensemble explicitly defines a variational approximation through the hyper-parameters of dropout (Srivastava et al., 2014) (used during training), allowing for straightforward sampling of model parameters. Another approach, SWA-Gaussian (Maddox et al., 2019), finds a Gaussian approximation based on the first two moments of stochastic gradient descent iterates. Unlike the deep ensemble approach, and similar to MC-dropout, this method allows for simple and efficient sampling but suffers from higher memory consumption. Even a diagonal Gaussian approximation

---

[1]In-depth comparisons of ensemble methods were conducted in Ovadia et al. (2019); Ashukha et al. (2020)

requires twice the memory of a standard network. There also exists alternative memory and/or compute efficient ensemble approaches such as BatchEnsembles (Wen et al., 2020) and MIMO (Havasi et al., 2021). While the former approach is parameter efficient it requires multiple forward passes at test time similar to MC ensembles. The latter avoids this issue by the use of independent subnetworks within a single deep model leading to both efficient training and testing in terms of computational and memory costs. However, MIMO still requires multiple output heads at test time; this is an issue when scaling to large scale classification tasks where the output layer has tens or hundreds of thousands classes.

Given an ensemble, the goal is to estimate and decompose the predictive uncertainty. First, the entropy of the predictive distribution $P(y|\boldsymbol{x}^*, \mathcal{D})$ can be seen as a measure of total uncertainty. Second, this can be decomposed (Depeweg et al., 2018; Kendall & Gal, 2017) as:

$$\underbrace{\mathcal{H}\big[P(y|\boldsymbol{x}^*, \mathcal{D})\big]}_{\text{Total Uncertainty}} = \underbrace{\mathcal{I}\big[y, \boldsymbol{\theta}|\boldsymbol{x}^*, \mathcal{D}\big]}_{\text{Knowledge Uncertainty}} + \underbrace{\mathbb{E}_{\text{p}(\boldsymbol{\theta}|\mathcal{D})}\big[\mathcal{H}[P(y|\boldsymbol{x}^*, \boldsymbol{\theta})]\big]}_{\text{Data Uncertainty}} \tag{3}$$

where $\mathcal{I}$ is mutual information and $\mathcal{H}$ represents entropy. This specific decomposition allows total uncertainty to be decomposed into separate estimates of knowledge and data uncertainty. Furthermore, the conditional mutual information can be rephrased as:

$$\mathcal{I}\big[y, \boldsymbol{\theta}|\boldsymbol{x}^*, \mathcal{D}\big] = \mathbb{E}_{\text{p}(\boldsymbol{\theta}|\mathcal{D})}\Big[\text{KL}\big(P(y|\boldsymbol{x}^*, \boldsymbol{\theta}) \,\|\, P(y|\boldsymbol{x}^*, \mathcal{D})\big)\Big] \tag{4}$$

For an in-domain sample $\boldsymbol{x}^*$ the mutual information should be low as appropriately trained models $P(y|\boldsymbol{x}^*, \boldsymbol{\theta})$ should be close to the predictive distribution. High predictive uncertainty will only occur if the input exists in a region of high data uncertainty, for example when an input has significant class overlap. When the input $\boldsymbol{x}^*$ is out-of-distribution of the training data, one should expect inconsistent, different, predictions $P(y|\boldsymbol{x}^*, \boldsymbol{\theta})$ leading to a much higher knowledge uncertainty estimate.

## 2.2 ENSEMBLE (DISTRIBUTION) DISTILLATION

Ensemble methods have generally shown superior performance on a range of tasks but suffer from being computationally expensive. To tackle this issue, a technique called *knowledge distillation* (KD) and its variants were developed for transferring the knowledge of an ensemble (teacher) into a single (student) model while maintaining good performance (Hinton et al., 2014; Kim & Rush, 2016; Guo et al., 2020). This is generally achieved by minimising the KL-divergence between the student prediction and the predictive distribution of the teacher ensemble. In essence, KD trains a new student model to predict the average prediction of its teacher model. However from the perspective of uncertainty estimation the student model no longer has any information about the diversity of various ensemble member predictions; it was only trained to model the average prediction. Hence, it is no longer possible to decompose the total uncertainty into different sources, only the total uncertainty can be obtained from the student. To tackle this issue *ensemble distribution distillation* (En2D) was developed (Malinin et al., 2020).

Let $\boldsymbol{\pi}$ signify a categorical distribution, that is $\pi_c = P(y = \omega_c|\boldsymbol{\pi})$. The goal is to directly model the space of categorical predictions $\{\boldsymbol{\pi}^{(m)} = \boldsymbol{f}(\boldsymbol{x}^*; \boldsymbol{\theta}^{(m)})\}_{m=1}^M$ made by the ensemble. In work developed by Malinin et al. (2020) this is done by letting a student model (with weights $\boldsymbol{\phi}$) predict the parameters of a Dirichlet, which is a continuous distribution over categorical distributions:

$$\text{p}(\boldsymbol{\pi}|\boldsymbol{x}^*, \boldsymbol{\phi}) = \text{Dir}(\boldsymbol{\pi}; \boldsymbol{\alpha}), \ \boldsymbol{\alpha} = \boldsymbol{f}(\boldsymbol{x}^*; \boldsymbol{\phi}) \tag{5}$$

The key idea in this concept is that we are not directly interested in the posterior $\text{p}(\boldsymbol{\theta}|\mathcal{D})$ but how predictions $\boldsymbol{\pi}$ for particular inputs behave when induced by this posterior. Therefore, it is possible to replace $\text{p}(\boldsymbol{\theta}|\mathcal{D})$ with a trained distribution $\text{p}(\boldsymbol{\pi}|\boldsymbol{x}^*, \boldsymbol{\phi})$. It is now necessary to train the student given the information from the teacher which is straightforwardly done using negative log-likelihood:

$$\mathcal{L}(\boldsymbol{\phi}) = -\frac{1}{M} \sum_{m=1}^M \ln \text{Dir}(\boldsymbol{\pi}^{(m)}; \boldsymbol{\alpha}) \tag{6}$$

A decomposable estimate of total uncertainty is then possible by using conditional mutual information between the class $y$ and prediction $\boldsymbol{\pi}$ instead of $\boldsymbol{\theta}$ Malinin & Gales (2018):

$$\underbrace{\mathcal{H}\big[P(y|\boldsymbol{x}^*, \boldsymbol{\phi})\big]}_{\text{Total Uncertainty}} = \underbrace{\mathcal{I}\big[y, \boldsymbol{\pi}|\boldsymbol{x}^*, \boldsymbol{\phi}\big]}_{\text{Knowledge Uncertainty}} + \underbrace{\mathbb{E}_{\text{p}(\boldsymbol{\pi}|\boldsymbol{x}^*, \boldsymbol{\phi})}\big[\mathcal{H}[P(y|\boldsymbol{\pi})]\big]}_{\text{Data Uncertainty}} \tag{7}$$

This decomposition has a similar interpretation to eq. (3). Using a Dirichlet model, these uncertainties can be found using a single forward pass, achieving a much higher level of efficiency compared to an ensemble. Assuming this distillation technique is successful, the distribution distilled student should be able to closely emulate the ensemble and be able to estimate similar high quality uncertainties on both ID and OOD data.

However, ensemble distribution distillation is only applicable and useful when the ensemble members are not overconfident and display diversity in their predictions—there is no need in capturing diversity when there is none. It is often the case that, for example, convolutional neural networks are over-parameterised, display severe overconfidence and can essentially achieve perfect training accuracy restricting the effectiveness of this method (Guo et al., 2017; Seo et al., 2019; Ryabinin et al., 2021). Furthermore, this method can only be used when an ensemble is available, leading to a high training cost.

## 3 SELF-DISTRIBUTION DISTILLATION

In this section we propose *self-distribution distillation* (S2D) for efficient training and uncertainty estimation, bypassing the need for a separate teacher ensemble. This combines:

- *parameter sharing*: allowing the teacher and student to share a common feature extraction base would accelerate training significantly, each will branch off and have their own head;
- *stochastic regularisation*: the teacher can generate multiple predictions efficiently by forward propagating an input through its head (with a stochastic regulariser) several times, emulating the behaviour of an ensemble;
- *distribution distillation*: while the teacher branch is trained on cross-entropy, the student is taught to predict a distribution over teacher predictions capturing the diversity compactly.

This process is summarised in Fig. 1. This approach can take many specific forms with regards to

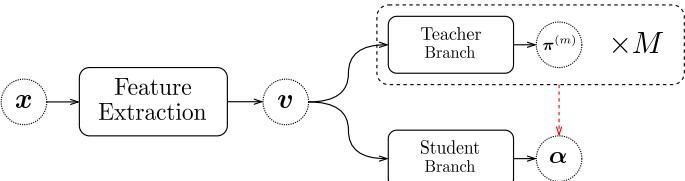

**Figure 1:** General structure of a self-distribution distilled model. $M$ stochastic teacher branch forward propagations are trained on cross-entropy and simultaneously distribution distilled to the student.

the type of feature extraction module, stochastic regulariser, teacher branch and student modelling choice. For example, the teacher could entail a much larger branch capturing complex patterns in the data, while the student could consist of a smaller branch used for compressing teacher knowledge into a more efficient form, at test time. On the other end, training efficiency can be achieved by forcing the teacher and student share the same branch parameters.

In this work we choose a particular model configuration that is highly efficient, shown in Fig. 2. The main functional difference between the teacher and the student branches is the use of logit values, $z$: for the teacher branch a probability is predicted; whereas the student uses the logits for a Dirichlet distribution. Furthermore the teacher uses stochastic regularisation techniques (SRTs) in generating multiple teacher predictions, analogous to an ensemble. In this work multiplicative Gaussian noise (Gaussian dropout) with unit mean and uniformly random standard deviation is used. This form

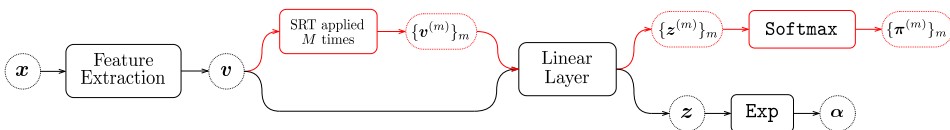

**Figure 2:** Dirichlet S2D model during training. Only the black part of the network is retained during prediction time, matching the behaviour of a standard model.

was chosen due to simplicity of sampling and possible ensemble diversity. There is a wide range of other choices regarding what SRTs to use, from Bernoulli dropout, additive Gaussian noise to deciding at which teacher branch layers this should be introduced. Furthermore, since the Dirichlet distribution has bounded ability to represent diverse ensemble predictions (Malinin et al., 2020), simply generating multiple teacher prediction by propagating through the last layer will not limit the ability of the model. To further improve the memory efficiency of the model, a single final linear layer shared by both student and teacher branches is used. This parameter sharing makes the S2D model efficient even when the number of classes is large, and does not use any more parameters compared to a standard model. Note any NN classifier can be cast into a self-distribution distillation format by inserting stochasticity prior to the final linear layer and can easily be combined with many other approaches such as MIMO (Havasi et al., 2021) and SWAG (Maddox et al., 2019).

This choice of integrating ensemble teacher training and distribution distillation into a single entity utilising parameter tying also serves as a regulariser (optimising two objectives using the same set of weights) and allows for inexpensive training. The only added training cost is from multiple forward passes through the final linear layer, a process which can be parallelised. Additionally, the restricted form of Fig. 2 brings some numerical stability. As noted by Malinin et al. (2020), optimising a student to predict a Dirichlet distribution can be unstable when there is a lack of common support between prediction and extremely sharp teacher outputs. However, note that teacher predictions are closely related to the expected student prediction:

$$\mathbb{E}_{\mathtt{Dir}(\boldsymbol{\pi};\boldsymbol{\alpha})}\big[\boldsymbol{\pi}\big] = \frac{\boldsymbol{\alpha}}{\alpha_0} = \mathtt{Softmax}(\boldsymbol{z}), \ \ \boldsymbol{\pi}^{(m)} = \mathtt{Softmax}(\boldsymbol{z}^{(m)}), \ \ \boldsymbol{z} = \mathbb{E}\big[\boldsymbol{z}^{(m)}\big] \tag{8}$$

leading to increased common support. Additionally, stochasticity in the teacher forces the outputs to have some diversity, mildly limiting overconfidence. Now we train the teacher branch using cross-entropy, and simultaneously, use the teacher predictions to train the student branch. Let the weights of this model be denoted by $\phi$ and say we have some input-target pair $(\boldsymbol{x}, y)$. The teacher loss (for a single sample) is then:

$$\mathcal{L}_{\mathtt{th}}(\boldsymbol{\phi}) = -\frac{1}{M} \sum_{m=1}^{M} \sum_{c} \delta(y, \omega_c) \ln \pi_c^{(m)} \tag{9}$$

where $\delta$ is the indicator function. The student branch could be trained using log-likelihood as in eq. (6) but it has been found that this approach could be unstable (Fathullah et al., 2021; Ryabinin et al., 2021). Instead we use the teacher categorical predictions in estimating a proxy teacher Dirichlet $\tilde{\boldsymbol{\alpha}}$ using maximum log-likelihood. The resulting student loss is KL-divergence based:

$$\mathcal{L}_{\mathtt{st}}(\boldsymbol{\phi}) = \mathtt{KL}\Big(\mathtt{Dir}(\boldsymbol{\pi}; \tilde{\boldsymbol{\alpha}}) \, \Big\| \, \mathtt{Dir}(\boldsymbol{\pi}; \boldsymbol{\alpha})\Big), \ \ \tilde{\boldsymbol{\alpha}} = \arg\max_{\hat{\boldsymbol{\alpha}}} \sum_m \ln \mathtt{Dir}(\boldsymbol{\pi}^{(m)}; \hat{\boldsymbol{\alpha}}) \tag{10}$$

The proxy Dirichlet is estimated using a numerical approach developed by Minka (2000). The overall training loss becomes $\mathcal{L}(\boldsymbol{\phi}) = \mathcal{L}_{\mathtt{th}}(\boldsymbol{\phi}) + \mu \mathcal{L}_{\mathtt{st}}(\boldsymbol{\phi})$ with a small constant $\mu$.

Deep learning models often overfit on training data leading to less informative outputs. To alleviate these issues we integrate temperature scaling in the student branch loss. While training the teacher branch predictions on cross-entropy we temperature scale the same predictions and use the resulting ones in estimating a proxy teacher Dirichlet. The student branch will repeatedly be taught to predict a smoother/wider Dirichlet distribution, while the teacher branch's objective is to maximise the probability of the correct class resulting in a middle ground.

## 4 SELF-DISTRIBUTION DISTILLED ENSEMBLE APPROACHES

If computational resources during the training phase are not constrained it would open up the possibility for self-distribution distilled ensembles and various hierarchical distillation approaches of such models. First it can be noted that the ensemble generation methods mentioned in previous sections can easily be used with the S2D models in the previous section. The predictive distribution of such an ensemble would take the following form:

$$\mathtt{P}(y = \omega_c | \boldsymbol{x}^*, \mathcal{D}) = \iint \mathtt{P}(y = \omega_c | \boldsymbol{\pi}) \mathtt{p}(\boldsymbol{\pi} | \boldsymbol{x}^*, \boldsymbol{\phi}) \mathtt{p}(\boldsymbol{\phi} | \mathcal{D}) \mathrm{d}\boldsymbol{\pi} \mathrm{d}\boldsymbol{\phi} \approx \frac{1}{M} \sum_{m=1}^{M} \frac{\alpha_c^{(m)}}{\alpha_0^{(m)}} \tag{11}$$

Furthermore, an ensemble of Dirichlet models can be used to estimate similar uncertainty measures as previously described:

$$\mathcal{H}\big[\mathrm{P}(y|\boldsymbol{x}^*,\mathcal{D})\big] = \mathcal{I}\big[y,\boldsymbol{\pi}|\boldsymbol{x}^*,\mathcal{D}\big] + \mathbb{E}_{\mathrm{p}(\boldsymbol{\phi}|\mathcal{D})}\Big[\mathbb{E}_{\mathrm{p}(\boldsymbol{\pi}|\boldsymbol{x}^*,\boldsymbol{\phi})}\big[\mathcal{H}[\mathrm{P}(y|\boldsymbol{\pi})]\big]\Big] \tag{12}$$

This is a generalisation of eq. (7) since specific weights $\boldsymbol{\phi}$ have been replaced with conditioning on the dataset $\mathcal{D}$. Computing these uncertainties requires only a few modifications compared to the standard ensemble in eq. (3).

Next, the most natural step is to transfer the knowledge of an S2D (Dirichlet) ensemble into a single model. A choice needs to be made regarding the hierarchy of student modelling: should the student predict a categorical[2], Dirichlet, or a distribution over Dirichlets—hereby given the family name *hierarchical distribution distillation* (H2D). Initially we start by training a student model to predict a single Dirichlet identical to eq. (5). However, since the S2D ensemble provides, for an input $\boldsymbol{x}^*$, a set of Dirichlets $\{\boldsymbol{\alpha}^{(m)} = \boldsymbol{f}(\boldsymbol{x}^*;\boldsymbol{\phi}^{(m)})\}_{m=1}^{M}$ a modified distillation criterion is needed:

$$\mathcal{L}(\boldsymbol{\phi}) = \frac{1}{M}\sum_{m=1}^{M}\mathtt{KL}\Big(\mathtt{Dir}(\boldsymbol{\pi};\boldsymbol{\alpha}^{(m)}) \,\Big\|\, \mathtt{Dir}(\boldsymbol{\pi};\boldsymbol{\alpha})\Big); \quad \boldsymbol{\alpha} = \boldsymbol{f}(\boldsymbol{x}^*;\boldsymbol{\phi}) \tag{13}$$

This KL-divergence based loss also allows the reverse KL criterion to be used (Malinin & Gales, 2019) if desired. One criticism of this form of model, Dirichlet H2D (H2D-Dir), is that the diversity across ensemble members is lost, similar to the drawback in standard distillation. Therefore, we seek a distribution over Dirichlets to capture this higher level of diversity.

To model the space of Dirichlets we need to define a distribution over the parameters. Here we are faced with a choice: (1) model the parameters $\boldsymbol{\alpha} \in \mathbb{R}_+^K$ directly (restricted to the non-negative real space) or (2) apply a transformation to simplify the modelling. Here a logarithmic transformation $\boldsymbol{z} = \ln\boldsymbol{\alpha} \in \mathbb{R}^K$ is applied and a simple distribution over the Dirichlet parameters, a diagonal Gaussian, to be used (see Appendix C for a justification for this modelling choice). With these building blocks, the goal of H2D is to train a student model, with weights $\boldsymbol{\lambda}$, predict the parameters of a diagonal Gaussian $(\boldsymbol{\mu},\boldsymbol{\sigma})$ (H2D-Gauss):

$$\mathrm{p}(\ln\boldsymbol{\alpha}|\boldsymbol{x}^*,\boldsymbol{\lambda}) = \mathcal{N}(\ln\boldsymbol{\alpha};\boldsymbol{\mu},\boldsymbol{\sigma}^2) = \prod_{c=1}^{K}\mathcal{N}(\ln\alpha_c;\mu_c,\sigma_c^2); \quad \boldsymbol{\mu},\boldsymbol{\sigma} = \boldsymbol{f}(\boldsymbol{x}^*;\boldsymbol{\lambda}) \tag{14}$$

By sampling from this Gaussian, one can obtain multiple Dirichlet distributions similar to, but cheaper than, an S2D ensemble. Clearly, the flexibility of such a model can easily be extended by allowing the model to predict a fully specified covariance, however due to computational tractability only diagonal covariance models are used in this work. A secondary head is required for such a model, see Fig. 3. In a similar fashion to previous approaches, this model can be trained using

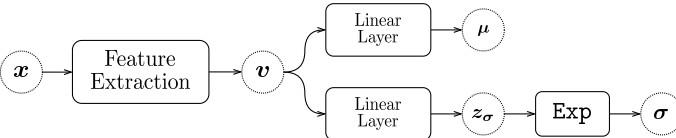

**Figure 3:** The architecture of a diagonal H2D-Gauss student.

negative log-likelihood or by estimating a proxy teacher Gaussian and use KL-divergence. In this work we have adopted the proxy approach, see Appendix A.1 for details

## 5 EXPERIMENTAL EVALUATION

This section investigates the self-distribution distillation approach on classifying image data. First, this approach is compared to standard trained models and established ensemble based methods (deep ensembles and MC-dropout) as well as the diagonal version of SWAG (SWAG-Diag) and MIMO. Second, self-distribution distillation is combined with all above mentioned approaches. Finally, knowledge distillation is compared to hierarchical distribution distillation of Dirichlet ensembles.

This comparison is based on two sets of experiments. The first set compares the performance of all baselines and proposed models in terms of image classification performance and calibration on

---

[2]Since transferring knowledge from a Dirichlet ensemble into a student predicting a categorical critically loses information about diversity, this method will not be investigated.

CIFAR-100 (Krizhevsky & Hinton, 2009) without (C100) and with (C100+) data augmentation. The second set of experiments compares the out-of-distribution (OOD) detection performance using the LSUN (Yu et al., 2015), Tiny ImageNet (CS231N, 2017) and SVHN (Netzer et al., 2011) datasets.

All experiments are based on training the DenseNet-BC (k = 12) with a depth of 100 (Huang et al., 2017). For ensemble generation methods $M = 5$ models were sampled (in the case of MC-dropout ensembles and SWAG) or trained (in the case of deep ensembles). For MIMO we use two output heads ($M = 2$) due to limited capacity in the chosen model (Havasi et al., 2021). Note that for this choice of model it was not possible to use ensemble distribution distillation since DenseNet-BC models display high confidence on the training data of CIFAR-100 causing instability in distillation. All single model training runs were repeated 5 times; mean $\pm$ 2 standard deviations are reported. The experimental setup and additional experiments are described in Appendix A-D.

## 5.1 CIFAR-100 CLASSIFICATION PERFORMANCE EXPERIMENTS

The first batch of experiments show the classification performance using a range of metrics such as accuracy, negative log-likelihood (NLL) and expected calibration error (ECE), see Table 1. Perhaps the most noteworthy result is the improvement in all metrics and datasets of a self-distribution distilled model compared to its standard counterpart. The improvement is more than 2 standard deviations. A similar picture can be observed for the S2D versions of SWAG-Diag and MC-dropout which, without any notable cost of training and inference, improve upon their equivalent standard counterparts in all metrics notably. Regarding MIMO a small gain can still be observed when switching to the self-distribution distillation framework but this boost is smaller. Finally for the deep ensemble approach, the S2D version only shows a marginal improvement in accuracy and NLL but a notable increase in ECE. In fact, it is observed that ensembling standard and S2D models reduces and increases ECE respectively. This trend is associated with the level of ensemble calibration. Unlike a standard deep ensemble, the members of the S2D counterpart are close to being calibrated, displaying little to no overconfidence. Ensembling these calibrated models lead to under-confident average predictions hence, the increased calibration error. Note, calibration error and negative log-likelihood can easily be reduced post-training by temperature scaling predictions.

The next set of comparisons regard distilled models, the final block of Table 1. As expected they all perform in between the performance of an individual model and deep ensemble. While standard ensemble distillation (knowledge distillation) was found to achieve better accuracy than other distillation methods, this success was highly dependent on the value of temperature scaling used. A sub-optimal choice of temperature can drastically reduce performance. On the other hand, when distilling an S2D ensemble, no additional hyper-parameters are needed. We observe that while both

**Table 1:** Test performance ($\pm$ 2 std) and train/test cost. Dropout regularisation was only used for C100. Inference times (per input) were estimated using an NVIDIA V100 GPU. *SWAG inference speeds do not take into account the time to update batch norm statistics.

| Dataset | C100 | | | C100+ | | | Computational Cost | |
|---|---|---|---|---|---|---|---|---|
| Model | Acc | NLL | %ECE | Acc | NLL | %ECE | Params | Inference |
| Individual | 74.6 $\pm$ 0.5 | 1.11 $\pm$ 0.07 | 11.95 $\pm$ 1.65 | 77.5 $\pm$ 0.2 | 1.01 $\pm$ 0.14 | 10.84 $\pm$ 2.32 | 0.80M | 2.3ms |
| S2D Individual | **75.7** $\pm$ 0.5 | **0.87** $\pm$ 0.02 | **2.54** $\pm$ 1.11 | **78.1** $\pm$ 0.4 | **0.81** $\pm$ 0.03 | **4.35** $\pm$ 1.23 | | |
| MIMO | 75.2 $\pm$ 0.6 | 1.05 $\pm$ 0.13 | 10.51 $\pm$ 2.75 | 77.6 $\pm$ 0.7 | 0.89 $\pm$ 0.18 | 8.23 $\pm$ 3.90 | 0.83M | 2.3ms |
| S2D MIMO | **75.4** $\pm$ 0.1 | **0.90** $\pm$ 0.08 | **5.77** $\pm$ 1.63 | **78.1** $\pm$ 0.6 | **0.80** $\pm$ 0.07 | **4.07** $\pm$ 0.43 | | |
| SWAG-Diag | 74.8 $\pm$ 1.0 | 1.08 $\pm$ 0.05 | 10.73 $\pm$ 1.31 | 77.7 $\pm$ 0.9 | 0.98 $\pm$ 0.03 | 9.60 $\pm$ 3.25 | 1.60M | 11.6ms* |
| S2D SWAG-Diag | **75.9** $\pm$ 0.6 | **0.85** $\pm$ 0.03 | **3.87** $\pm$ 0.88 | **78.2** $\pm$ 1.3 | **0.79** $\pm$ 0.07 | **3.65** $\pm$ 0.62 | | |
| MC ensemble | 75.6 $\pm$ 0.9 | 0.94 $\pm$ 0.04 | 6.67 $\pm$ 1.18 | - | - | - | 0.80M | 11.5ms |
| S2D MC ensemble | **76.6** $\pm$ 0.4 | **0.83** $\pm$ 0.02 | **2.57** $\pm$ 0.58 | - | - | - | | |
| Deep ensemble | 79.3 | 0.76 | **1.44** | **82.1** | 0.66 | **1.61** | 4.00M | 11.5ms |
| S2D Deep ensemble | **79.7** | **0.73** | 5.48 | **82.1** | **0.64** | 3.79 | | |
| EnD | **77.9** | 0.91 | 10.36 | **81.2** | 0.81 | 9.51 | 0.80M | 2.3ms |
| H2D-Dir | 77.7 | 0.84 | 3.24 | 80.9 | 0.71 | 3.42 | | |
| H2D-Gauss | 77.5 | **0.77** | **1.39** | 80.5 | **0.68** | **2.41** | 0.83M | 2.4ms |

H2D-Dir and H2D-Gauss obtained a higher NLL they also achieved better calibration than their S2D ensemble teacher. Lastly, one can observe that H2D-Dir and H2D-Gauss both outperform the standard SWAG-Diag and MC-dropout ensemble using only a single forward pass. Although these distilled models involve an expensive training phase (a teacher ensemble is required) they are able to, at test time, achieve much higher computational efficiency and estimate and decompose uncertainty.

## 5.2 OUT-OF-DISTRIBUTION DETECTION EXPERIMENTS

The second batch of experiments investigate the out-of-distribution detection performance of models. The goal is to differentiate between two types of data, negative in-distribution (ID, sampled from the same source as the training data) and positive out-of-distribution (OOD) data.

In all experiments the models were trained on C100. The ID data was always set to the test set of C100 and OOD data was the test set of LSUN/TIM/SVHN. Both LSUN and TIM examples had to be resized or randomly cropped as preprocessing before being fed to the model. The detection was done using four uncertainty estimates: confidence, total uncertainty (TU), data or aleatoric uncertainty (DU) and knowledge or epistemic uncertainty (KU). Performance was measured using the threshold independent AUROC (Manning & Schütze, 1999) and AUPR (Fawcett, 2006) metrics. Due to limited space, some LSUN and TIM experiments have been moved to Appendix B.1.

First, there is not a single case in Tables 2 and 3 where an individual model, MIMO, SWAG-Diag or MC-dropout ensemble is able to outperform the detection performance of a single S2D model. This statement holds for all the analysed uncertainties apart from confidence where both MIMO and SWAG-Diag are insignificantly better. When comparing to a deep ensemble, the S2D model is outperformed in many cases. The general trend is that the ensemble is able to output marginally higher quality confidence and total uncertainty estimates in most datasets, but that S2D sometimes outperforms the ensemble when using data uncertainty (as in Table 3).

Interestingly, the MC ensemble seems to degrade the quality of confidence and total uncertainty when compared to its standard individual counterpart. However, since a MC-dropout ensemble can estimate data uncertainty, it is able to outperform the standard model overall. Similarly, the S2D MC ensemble generally has inferior detection performance compared to its single deterministic model equivalent. The only exception is in detecting SVHN where the ensemble has marginally better data uncertainty estimates. Regarding SWAG-Diag and MIMO they both gain from being cast into a self-distribution distillation viewpoint drastically increasing their detection performance without additional cost at inference.

Although the S2D deep ensemble, when compared to its vanilla counterpart, wasn't able to show any noticeable accuracy boost (on CIFAR-100) it does outperform in this detection task. The only case where the S2D ensemble was not able to outshine the vanilla ensemble is when both use knowl-

**Table 2:** OOD detection results (LSUN resize) trained on C100. **Best** in column and **best** overall.

| Model | OOD %AUROC | | | | OOD %AUPR | | | |
|---|---|---|---|---|---|---|---|---|
| | Conf. | TU | DU | KU | Conf. | TU | DU | KU |
| Individual | $77.3 \pm 0.9$ | $79.8 \pm 0.9$ | | | $74.2 \pm 1.1$ | $76.9 \pm 1.2$ | | |
| S2D Individual | $78.4 \pm 2.3$ | $80.7 \pm 3.2$ | $80.8 \pm 3.1$ | $80.0 \pm 4.2$ | $75.4 \pm 2.5$ | $78.3 \pm 3.5$ | $79.5 \pm 3.5$ | $75.5 \pm 3.8$ |
| MIMO | $78.5 \pm 1.2$ | $80.5 \pm 1.4$ | $80.6 \pm 1.4$ | $75.0 \pm 2.8$ | $75.0 \pm 1.4$ | $78.0 \pm 1.6$ | $78.1 \pm 1.6$ | $67.0 \pm 3.5$ |
| S2D MIMO | $80.6 \pm 4.1$ | $81.4 \pm 4.4$ | $81.4 \pm 4.4$ | $81.3 \pm 4.2$ | $76.6 \pm 5.2$ | $78.8 \pm 5.4$ | $80.3 \pm 5.4$ | $77.7 \pm 5.3$ |
| SWAG-Diag | $78.5 \pm 1.0$ | $80.5 \pm 1.2$ | $80.6 \pm 1.3$ | $75.2 \pm 0.8$ | $75.0 \pm 1.4$ | $78.1 \pm 1.7$ | $78.3 \pm 1.8$ | $67.1 \pm 1.0$ |
| S2D SWAG-Diag | $78.7 \pm 2.3$ | $80.9 \pm 2.8$ | $81.1 \pm 2.7$ | $80.9 \pm 3.8$ | $75.4 \pm 2.7$ | $78.4 \pm 3.6$ | $79.7 \pm 3.2$ | $76.2 \pm 4.1$ |
| MC ensemble | $76.6 \pm 0.8$ | $78.3 \pm 0.8$ | $78.9 \pm 0.8$ | $72.4 \pm 1.2$ | $72.2 \pm 1.0$ | $74.6 \pm 1.6$ | $75.6 \pm 1.7$ | $64.2 \pm 2.0$ |
| S2D MC ensemble | $77.7 \pm 0.9$ | $79.8 \pm 1.5$ | $80.5 \pm 1.1$ | $78.1 \pm 2.9$ | $73.7 \pm 1.0$ | $76.1 \pm 1.7$ | $78.6 \pm 1.3$ | $72.0 \pm 3.2$ |
| Deep ensemble | 81.1 | 82.9 | 83.4 | 79.2 | 77.7 | 80.4 | 81.2 | 73.6 |
| S2D Deep Ensemble | **82.4** | **84.8** | 85.0 | 83.5 | **79.5** | **82.5** | 83.9 | 78.7 |
| EnD | 79.4 | 81.0 | | | 75.8 | 78.2 | | |
| H2D-Dir | 80.3 | 83.2 | 83.4 | **86.4** | 77.9 | 81.9 | 81.9 | **83.4** |
| H2D-Gauss | 80.8 | 83.9 | **85.7** | 80.7 | 78.2 | 82.0 | **85.8** | 76.0 |

**Table 3:** OOD detection results (SVHN) trained on C100. **Best** in column and **best** overall.

| | | | | | | | | |
|---|---|---|---|---|---|---|---|---|
| Individual | $79.7 \pm 5.6$ | $81.8 \pm 6.0$ | | | $88.3 \pm 3.6$ | $89.6 \pm 3.9$ | | |
| S2D Individual | $83.0 \pm 2.9$ | $86.0 \pm 2.2$ | $87.7 \pm 2.2$ | $81.2 \pm 3.8$ | $90.6 \pm 1.7$ | $92.0 \pm 1.6$ | $94.4 \pm 1.1$ | $86.1 \pm 3.3$ |
| MIMO | $81.8 \pm 4.1$ | $84.3 \pm 4.5$ | $84.3 \pm 4.5$ | $80.9 \pm 5.3$ | $89.9 \pm 2.5$ | $91.4 \pm 2.8$ | $91.4 \pm 2.8$ | $88.2 \pm 3.1$ |
| S2D MIMO | $84.1 \pm 2.3$ | $87.2 \pm 2.1$ | $87.4 \pm 2.1$ | $83.7 \pm 1.8$ | $89.6 \pm 1.8$ | $92.9 \pm 1.6$ | $93.2 \pm 1.6$ | $90.4 \pm 1.3$ |
| SWAG-Diag | $81.4 \pm 3.0$ | $83.5 \pm 3.6$ | $83.5 \pm 3.4$ | $80.5 \pm 4.9$ | $89.2 \pm 2.6$ | $90.2 \pm 3.2$ | $90.2 \pm 3.1$ | $88.3 \pm 3.6$ |
| S2D SWAG-Diag | $83.2 \pm 2.7$ | $86.3 \pm 2.6$ | $87.7 \pm 2.5$ | $82.7 \pm 4.3$ | $90.7 \pm 1.7$ | $92.3 \pm 1.8$ | $94.3 \pm 1.4$ | $87.3 \pm 3.2$ |
| MC ensemble | $79.0 \pm 4.3$ | $81.6 \pm 4.7$ | $83.1 \pm 4.6$ | $68.3 \pm 3.0$ | $88.1 \pm 2.8$ | $89.3 \pm 3.3$ | $90.7 \pm 3.1$ | $77.4 \pm 1.8$ |
| S2D MC ensemble | $82.3 \pm 4.3$ | $85.9 \pm 4.1$ | $88.4 \pm 3.5$ | $79.7 \pm 6.1$ | $90.5 \pm 2.6$ | $92.1 \pm 2.7$ | $95.0 \pm 1.7$ | $85.4 \pm 4.2$ |
| Deep ensemble | 84.5 | 87.2 | 86.8 | 85.0 | 91.3 | 92.5 | 92.2 | **91.5** |
| S2D Deep ensemble | **86.5** | **89.9** | **91.7** | 85.1 | **92.6** | **94.1** | **96.2** | 88.4 |
| EnD | 78.0 | 79.8 | | | 87.0 | 87.9 | | |
| H2D-Dir | 84.6 | 88.4 | 88.5 | **87.6** | 91.7 | 93.6 | 91.7 | 90.6 |
| H2D-Gauss | 81.2 | 85.3 | 90.1 | 74.5 | 90.0 | 91.4 | 95.9 | 81.7 |

edge uncertainty to detect SVHN examples using the AUPR metric. Generally, S2D based systems outperform their standard counterparts.

Regarding distillation based approaches, it is observed that knowledge ensemble distillation, EnD, is able to outperform the standard model in all cases except SVHN detection, and in no case is able to reach the deep ensemble performance, which it was distilled from. On the other hand, both the H2D-Dir and H2D-Gauss models outperform the distilled model and are able to decompose predictive uncertainty. Specifically we discover that H2D-Dir is able to generate the highest quality knowledge uncertainty estimates in almost all cases, and is able to outperform its S2D ensemble teacher using this uncertainty. The H2D-Gauss model however, was not able to boast similar high quality knowledge uncertainty. Instead, this model displayed the generally best performing data uncertainty estimates, able to outperform the vanilla deep ensemble in all cases, and the S2D equivalent in all but SVHN detection.

## 6 CONCLUSION

Uncertainty estimation within deep learning is becoming increasingly importance, with deep ensembles being the standard for estimating various sources of uncertainty. However, ensembles suffer from significantly higher computational requirements. This work proposes *self-distribution distillation* (S2D), a novel collection of approaches for directly training models able to estimate and decompose predictive uncertainty, without explicitly training an ensemble, and can seamlessly be combined with other approaches. Additionally, if one is not resource restricted during the training phase, a novel approach, *hierarchical distribution distillation* (H2D), is described for transferring/distilling the knowledge of S2D style ensembles into a single flexible and robust student model. It is shown that S2D models are able to outperform standard models and rival MC ensembles on the CIFAR-100 test set. Additionally, S2D is able to estimate higher quality uncertainty estimates compared to standard models and MC ensembles and in most cases, able to better detect out-of-distribution images from the LSUN, SVHN and TIM datasets. Combination of S2D with other promising approaches such as MIMO and SWAG also show additional gains in accuracy and detection performance. S2D is also able to rival the deep ensemble in certain cases even though it only requires a single forward pass. Furthermore, S2D deep ensembles and H2D derived student models are shown to notably outperform the deep ensemble in almost all detection problems. These promising results show that the efficient self-distribution and novel hierarchical distribution distillation approaches have the potential to train robust uncertainty estimating models able to outperform deep ensembles. Future work should further investigate self-distribution distillation in other domains such as natural language processing and speech recognition. The need for more efficient uncertainty estimation is especially useful for these areas as they often utilise large-scale models. Furthermore, one could also analyse variations of S2D such as utilising less weight sharing, generating more diverse teacher predictions or changing the student modelling choices.

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

# A EXPERIMENTAL CONFIGURATION

**Table 4:** Description of datasets used in training and evaluating models.

| Dataset | Train | Test | Classes |
|---|---|---|---|
| CIFAR-100 | 50000 | 10000 | 100 |
| LSUN | - | 10000 | 10 |
| SVHN | - | 26032 | 10 |
| Tiny ImageNet | - | 10000 | 200 |

All models were trained on the CIFAR-100 dataset, with and without data augmentation. The augmentation scheme involves randomly mirroring and shifting images following He et al. (2016); Huang et al. (2016). Remaining datasets were used as out-of-distribution samples in the detection task.

All individual models, and ensemble members were based of off the DenseNet-BC ($k = 12$, 100 layers) architecture and trained according to Huang et al. (2017). SWAG-Diag was obtained by checkpointing the weights of the last 20 epochs with a reduced learning rate of $\eta = 1.0 \times 10^{-4}$. MIMO with two output heads was trained using the same setup as for the standard model. To keep training costs comparable to (S2D) individual models, no batch or input repetition was used (Havasi et al., 2021). Similarly all self-distribution distilled equivalents were trained with identical training recipes with the addition of a student loss ($\mu = 1.28 \times 10^{-4}$).

Regarding distilled based models, the EnD baseline was trained using negative log-likelihood using the average temperature scaled prediction of the teacher ensemble, with $T \in \{1.0, 2.0, 3.0, 4.0, 5.0\}$. For the hierarchical distribution distillation approaches the students were first initialised with the weights of an S2D model trained for 150 epochs, for increased stability. Thereafter, each student was trained using the appropriate H2D criteria with a significantly reduced learning rate. H2D-Dir was trained using $\eta = 5.0 \times 10^{-5}$ for an additional 150 epochs. H2D-Gauss required an initial learning rate of $\eta = 5.0 \times 10^{-3}$ which was reduced by a factor of 2 after 75 and 150 epochs. It was trained for 170 epochs. Additionally, uncertainties were computed by generating 50 samples from each Gaussian prediction, since this modelling choice does not result in closed form expressions.

## A.1 PROXY TARGET TRAINING

Since the use of negative log-likelihood can be unstable in training S2D and distilling H2D models we utilise proxy targets and KL-divergence. It has already been mentioned that the proxy target in S2D follows:

$$\tilde{\boldsymbol{\alpha}} = \arg\max_{\hat{\boldsymbol{\alpha}}} \sum_m \ln \mathtt{Dir}(\boldsymbol{\pi}^{(m)}; \hat{\boldsymbol{\alpha}}), \;\; \boldsymbol{\pi}^{(m)} = \mathtt{Softmax}(\boldsymbol{z}^{(m)}, T) \quad (15)$$

Each categorical prediction will be temperature scaled, with $T = 1.5$, to mitigate overconfident predictions. While H2D-Dir does not require any proxy targets, the Gaussian equivalent does. The proxy diagonal Gaussian, estimated according to maximum log-likelihood, has a closed-form expression:

$$\tilde{\boldsymbol{\mu}} = \frac{1}{M} \sum_{m=1}^{M} \ln \boldsymbol{\alpha}^{(m)}, \;\; \tilde{\boldsymbol{\sigma}}^2 = \frac{1}{M} \sum_{m=1}^{M} (\ln \boldsymbol{\alpha}^{(m)} - \tilde{\boldsymbol{\mu}})^2 \quad (16)$$

where $\boldsymbol{v}^2 = \boldsymbol{v} \odot \boldsymbol{v}$ represents an element-wise multiplication. This is then used in a KL-divergence based loss, training the student with prediction $\boldsymbol{\mu}, \boldsymbol{\sigma}$ according to:

$$\mathtt{KL}\Big(\mathcal{N}(\boldsymbol{z}; \tilde{\boldsymbol{\mu}}, \tilde{\boldsymbol{\sigma}}^2) \,\Big\|\, \mathcal{N}(\boldsymbol{z}; \boldsymbol{\mu}, \boldsymbol{\sigma}^2)\Big) = \sum_{c=1}^{K} \ln\left(\frac{\sigma_c}{\tilde{\sigma}_c}\right) + \frac{\tilde{\sigma}_c^2 + (\mu_c - \tilde{\mu}_c)^2}{2\sigma_c^2} - \frac{1}{2} \quad (17)$$

Note however, that the proxy targets are detached from any back gradient propagation calculations. This is to simulate typical teacher-student knowledge transfer where teacher weights are kept fixed during student training.

# B  OUT-OF-DISTRIBUTION DETECTION

This section covers remaining out-of-distribution detection experiments. First, we cover the LSUN and Tiny ImageNet detection problem for all models considered in section 5.2. Thereafter, additional experiments will be run on ensembles of various sizes. This is to investigate if the low quality of knowledge uncertainty estimates is caused by a limited number of ensemble members.

## B.1  TINY IMAGENET EXPERIMENTS

Similar to the results section 5.2 the S2D Deep ensemble and H2D-Gauss outperformed all other models, see Table 6 and 7. The only exception is the use of confidence on resized TIM with the AU-ROC metric where the Deep ensemble marginally outperforms the S2D equivalent. However, unlike previous results, knowledge uncertainty seems to perform on par with or outperform confidence. The one exception is the MC ensemble.

**Table 5:** OOD detection results (LSUN random crop) trained on C100. **Best** in column and **best** overall.

| Model | OOD %AUROC | | | | OOD %AUPR | | | |
|---|---|---|---|---|---|---|---|---|
| | Conf. | TU | DU | KU | Conf. | TU | DU | KU |
| Individual | $83.2 \pm 2.1$ | $85.7 \pm 4.5$ | | | $79.4 \pm 5.6$ | $83.0 \pm 5.9$ | | |
| S2D Individual | $85.4 \pm 4.5$ | $88.9 \pm 4.1$ | $90.3 \pm 4.0$ | $84.1 \pm 4.8$ | $81.9 \pm 6.2$ | $86.6 \pm 5.7$ | $90.3 \pm 5.0$ | $76.0 \pm 5.1$ |
| MIMO | $83.3 \pm 3.9$ | $86.2 \pm 4.2$ | $86.3 \pm 4.3$ | $80.9 \pm 1.6$ | $79.6 \pm 6.6$ | $83.8 \pm 6.8$ | $83.8 \pm 6.9$ | $72.4 \pm 3.7$ |
| S2D MIMO | $85.8 \pm 2.5$ | $89.5 \pm 2.8$ | $90.7 \pm 2.8$ | $85.5 \pm 2.8$ | $78.0 \pm 3.4$ | $84.8 \pm 3.5$ | $89.4 \pm 3.4$ | $75.2 \pm 3.3$ |
| SWAG-Diag | $84.3 \pm 2.8$ | $87.1 \pm 3.1$ | $87.1 \pm 3.1$ | $80.8 \pm 7.2$ | $80.8 \pm 4.0$ | $84.5 \pm 3.8$ | $84.6 \pm 3.8$ | $73.4 \pm 14.2$ |
| S2D SWAG-Diag | $85.6 \pm 2.7$ | $89.1 \pm 2.5$ | $90.4 \pm 2.5$ | $85.3 \pm 3.0$ | $81.8 \pm 4.0$ | $86.5 \pm 3.6$ | $90.2 \pm 3.4$ | $76.4 \pm 3.6$ |
| MC ensemble | $81.0 \pm 3.5$ | $84.4 \pm 4.0$ | $86.4 \pm 3.8$ | $63.0 \pm 4.0$ | $77.0 \pm 3.6$ | $81.7 \pm 4.0$ | $84.9 \pm 4.0$ | $53.1 \pm 3.1$ |
| S2D MC ensemble | $83.3 \pm 2.3$ | $86.9 \pm 3.2$ | $90.0 \pm 2.5$ | $77.7 \pm 5.3$ | $79.3 \pm 3.2$ | $83.8 \pm 4.3$ | $90.1 \pm 3.2$ | $69.8 \pm 4.8$ |
| Deep ensemble | 85.9 | 89.1 | 90.9 | 80.4 | 82.0 | 86.3 | 89.1 | 72.5 |
| S2D Deep ensemble | 86.8 | 90.5 | 93.7 | 81.5 | **83.0** | 87.9 | 93.9 | 73.4 |
| EnD | 84.7 | 87.4 | | | 81.1 | 84.9 | | |
| H2D-Dir | 85.3 | 88.9 | 88.8 | **91.7** | 82.5 | 87.4 | 87.6 | **87.1** |
| H2D-Gauss | **86.9** | **90.6** | **95.1** | 76.0 | 82.9 | **88.0** | **95.7** | 67.0 |

**Table 6:** OOD detection results (TIM resize) trained on C100. **Best** in column and **best** overall.

| Model | OOD %AUROC | | | | OOD %AUPR | | | |
|---|---|---|---|---|---|---|---|---|
| Individual | $77.6 \pm 0.7$ | $79.5 \pm 0.7$ | | | $74.2 \pm 0.7$ | $77.1 \pm 0.9$ | | |
| S2D Individual | $78.0 \pm 0.8$ | $80.1 \pm 0.7$ | $79.6 \pm 0.8$ | $78.1 \pm 0.4$ | $75.3 \pm 0.9$ | $77.7 \pm 0.9$ | $76.6 \pm 1.2$ | $76.3 \pm 0.5$ |
| MIMO | $78.1 \pm 0.4$ | $79.9 \pm 0.7$ | $79.9 \pm 0.8$ | $76.3 \pm 1.5$ | $74.6 \pm 1.0$ | $77.3 \pm 1.3$ | $77.4 \pm 1.3$ | $69.6 \pm 2.0$ |
| S2D MIMO | $80.1 \pm 1.2$ | $80.7 \pm 1.2$ | $80.7 \pm 1.2$ | $80.4 \pm 1.2$ | $77.3 \pm 1.6$ | $77.8 \pm 1.6$ | $77.7 \pm 1.5$ | $77.5 \pm 1.6$ |
| SWAG-Diag | $77.7 \pm 0.7$ | $79.6 \pm 0.6$ | $79.6 \pm 0.6$ | $76.4 \pm 0.7$ | $74.2 \pm 0.8$ | $77.0 \pm 0.8$ | $77.1 \pm 0.8$ | $70.0 \pm 0.7$ |
| S2D SWAG-Diag | $78.6 \pm 0.7$ | $80.5 \pm 0.6$ | $80.1 \pm 0.7$ | $79.2 \pm 0.5$ | $75.6 \pm 0.9$ | $78.1 \pm 1.1$ | $77.1 \pm 1.0$ | $76.5 \pm 0.9$ |
| MC ensemble | $78.5 \pm 0.5$ | $80.6 \pm 0.3$ | $80.8 \pm 0.4$ | $76.6 \pm 0.6$ | $75.2 \pm 0.5$ | $78.1 \pm 0.6$ | $78.4 \pm 0.5$ | $70.9 \pm 1.1$ |
| S2D MC ensemble | $79.3 \pm 0.5$ | $81.1 \pm 0.5$ | $81.1 \pm 0.5$ | $80.4 \pm 0.6$ | $76.4 \pm 0.7$ | $78.5 \pm 0.8$ | $78.1 \pm 1.0$ | $77.1 \pm 0.7$ |
| Deep ensemble | **81.7** | 83.6 | 83.5 | 81.0 | 78.9 | 81.6 | 81.5 | 76.6 |
| S2D Deep Ensemble | 81.5 | **84.2** | 82.8 | 82.8 | **79.1** | **82.0** | 79.9 | 80.0 |
| EnD | 78.7 | 80.4 | | | 75.4 | 78.0 | | |
| H2D-Dir | 77.3 | 79.8 | 79.6 | 81.6 | 74.5 | 77.9 | 77.7 | 79.2 |
| H2D-Gauss | 80.5 | 82.6 | **83.7** | **82.8** | 78.8 | 81.4 | **82.5** | **80.1** |

**Table 7:** OOD detection results (TIM random crop) trained on C100. **Best** in column and **best** overall.

| | | | | | | | | |
|---|---|---|---|---|---|---|---|---|
| Individual | 76.7 ±4.1 | 79.2 ±4.2 | | | 74.7 ±3.6 | 78.5 ±3.8 | | |
| S2D Individual | 80.2 ±5.9 | 85.4 ±6.2 | 84.5 ±5.9 | 86.4 ±6.3 | 79.3 ±6.3 | 83.3 ±6.7 | 81.9 ±6.6 | 83.1 ±6.7 |
| MIMO | 79.4 ±4.8 | 81.9 ±5.3 | 81.9 ±5.3 | 79.8 ±4.6 | 77.1 ±4.8 | 80.9 ±5.2 | 80.8 ±5.3 | 74.9 ±8.1 |
| S2D MIMO | 80.3 ±8.6 | 86.5 ±8.5 | 86.5 ±8.5 | 86.9 ±8.6 | 80.0 ±6.5 | 82.9 ±6.4 | 83.0 ±6.4 | 84.9 ±6.5 |
| SWAG-Diag | 78.4 ±3.5 | 80.9 ±3.7 | 80.9 ±4.0 | 78.6 ±2.0 | 76.0 ±3.3 | 79.8 ±3.4 | 79.7 ±3.7 | 73.7 ±3.5 |
| S2D SWAG-Diag | 80.5 ±6.0 | 84.8 ±6.5 | 83.8 ±6.3 | 86.6 ±6.6 | 79.4 ±5.5 | 83.4 ±6.1 | 81.8 ±6.2 | 83.4 ±6.0 |
| MC ensemble | 75.8 ±4.5 | 78.8 ±4.8 | 79.7 ±4.9 | 69.3 ±3.7 | 74.3 ±4.0 | 78.5 ±4.3 | 80.0 ±4.3 | 60.8 ±3.7 |
| S2D MC ensemble | 78.8 ±6.3 | 82.1 ±6.4 | 82.6 ±6.5 | 82.0 ±6.1 | 77.1 ±5.2 | 81.1 ±5.1 | 81.8 ±5.1 | 79.8 ±4.9 |
| Deep ensemble | 80.9 | 84.2 | 83.5 | 82.3 | 79.3 | 83.9 | 83.2 | 79.8 |
| S2D Deep ensemble | **84.8** | **88.5** | 86.4 | **89.7** | **82.8** | **87.3** | 84.4 | **87.7** |
| EnD | 72.7 | 74.8 | | | 71.4 | 75.0 | | |
| H2D-Dir | 74.7 | 78.2 | 77.9 | 84.2 | 73.2 | 77.7 | 77.5 | 81.7 |
| H2D-Gauss | 83.2 | 88.0 | **88.0** | 88.5 | 81.0 | 86.0 | **87.2** | 84.1 |

## B.2 Ensemble Size Experiments

Knowledge uncertainty was found to have underwhelming performance (especially for MC and Deep ensembles) and did not show similar trends to prior work (Malinin & Gales, 2018; 2021; Malinin et al., 2020). To possibly mitigate this, the ensemble size was increased as a smaller number of models could lead to inaccurate measures of diversity and knowledge uncertainty. Results are compiled in Tables 8-13.

Performance on the CIFAR-100 test set is shown in Table 8. Increasing the ensemble size leads to improved accuracy and lower negative log-likelihoods as would be expected. The MC ensemble also becomes better calibrated. The Deep ensemble on the other hand has increasing calibration error with the number of members. This is due to the ensemble prediction becoming under-confident when averaging over a large number of members.

Out-of-distribution detection performance on LSUN, SVHN and TIM are compiled in Tables 9-13. Although the MC ensemble enjoys improved accuracy when increased in size, it seems to remain relatively unaffected in terms of OOD detection using any uncertainty metric. In detecting LSUN using random crops, the performance of KU interestingly deteriorates notably. Overall this points to MC ensembles' lacking ability in utilising new information from additional ensemble member draws/samples for better uncertainty estimation. Regarding the Deep ensemble, it generally improves with increasing size with any metric, however with diminishing returns. In this case all uncertainties improve with ensemble size, not only knowledge uncertainty. Therefore it seems that the cause for confidence, total and data outperforming knowledge uncertainty is not due to the ensemble size being limited to five members.

**Table 8:** Test performance of various ensembles and sizes (± 2 std). All models are trained on C100.

| Ensemble Type | Ensemble Size (M) | Acc. | NLL | %ECE |
|---|---|---|---|---|
| MC | 5 | 75.6 ±0.9 | 0.94 ±0.04 | 6.67 ±1.18 |
| | 10 | 75.8 ±0.9 | 0.92 ±0.04 | 6.11 ±1.11 |
| | 20 | 76.0 ±1.0 | 0.91 ±0.04 | 5.81 ±1.12 |
| Deep | 5 | 79.3 | 0.76 | 1.44 |
| | 10 | 80.1 | 0.71 | 1.91 |
| | 20 | 80.3 | 0.68 | 2.19 |

**Table 9:** OOD detection results (LSUN resize) trained on C100.

| Type | M | OOD %AUROC | | | | OOD %AUPR | | | |
|------|---|------|------|------|------|------|------|------|------|
| | | Conf. | TU | DU | KU | Conf. | TU | DU | KU |
| MC | 5 | $76.6 \pm 0.8$ | $78.3 \pm 0.8$ | $78.9 \pm 0.8$ | $72.4 \pm 1.2$ | $72.2 \pm 1.0$ | $74.6 \pm 1.6$ | $75.6 \pm 1.7$ | $64.2 \pm 2.0$ |
| | 10 | $76.7 \pm 0.6$ | $78.3 \pm 0.8$ | $79.1 \pm 0.9$ | $72.6 \pm 1.2$ | $72.3 \pm 1.1$ | $74.6 \pm 1.6$ | $75.9 \pm 1.7$ | $64.3 \pm 2.0$ |
| | 20 | $76.8 \pm 0.7$ | $78.4 \pm 0.8$ | $79.2 \pm 0.8$ | $72.7 \pm 1.3$ | $72.4 \pm 1.2$ | $74.6 \pm 1.6$ | $76.0 \pm 1.7$ | $64.3 \pm 2.3$ |
| Deep | 5 | 81.1 | 82.9 | 83.4 | 79.2 | 77.7 | 80.4 | 81.2 | 73.6 |
| | 10 | 82.0 | 83.9 | 84.8 | 80.3 | 79.1 | 81.8 | 83.4 | 74.9 |
| | 20 | 82.2 | 84.0 | 85.1 | 80.9 | 79.4 | 81.8 | 83.6 | 75.7 |

**Table 10:** OOD detection results (LSUN random crop) trained on C100.

| Type | M | Conf. | TU | DU | KU | Conf. | TU | DU | KU |
|------|---|------|------|------|------|------|------|------|------|
| MC | 5 | $81.0 \pm 3.5$ | $84.4 \pm 4.0$ | $86.4 \pm 3.8$ | $63.0 \pm 4.0$ | $77.0 \pm 3.6$ | $81.7 \pm 4.0$ | $84.9 \pm 4.0$ | $53.1 \pm 3.1$ |
| | 10 | $81.0 \pm 3.5$ | $84.4 \pm 3.9$ | $86.7 \pm 3.7$ | $61.6 \pm 3.9$ | $77.0 \pm 3.7$ | $81.8 \pm 4.0$ | $85.4 \pm 4.0$ | $52.2 \pm 3.0$ |
| | 20 | $80.8 \pm 3.7$ | $84.1 \pm 4.1$ | $86.6 \pm 3.9$ | $60.9 \pm 4.0$ | $76.7 \pm 3.9$ | $81.3 \pm 4.2$ | $85.3 \pm 4.2$ | $51.7 \pm 3.0$ |
| Deep | 5 | 85.9 | 89.1 | 90.9 | 80.4 | 82.0 | 86.3 | 89.1 | 72.5 |
| | 10 | 85.7 | 89.3 | 91.3 | 81.3 | 81.8 | 86.4 | 89.9 | 73.1 |
| | 20 | 86.2 | 89.8 | 92.2 | 82.0 | 82.1 | 86.8 | 91.0 | 73.1 |

**Table 11:** OOD detection results (SVHN) trained on C100.

| Type | M | Conf. | TU | DU | KU | Conf. | TU | DU | KU |
|------|---|------|------|------|------|------|------|------|------|
| MC | 5 | $79.0 \pm 4.3$ | $81.6 \pm 4.7$ | $83.1 \pm 4.6$ | $68.3 \pm 3.0$ | $88.1 \pm 2.8$ | $89.3 \pm 3.3$ | $90.7 \pm 3.1$ | $77.4 \pm 1.8$ |
| | 10 | $78.9 \pm 4.4$ | $81.5 \pm 4.7$ | $83.3 \pm 4.7$ | $67.5 \pm 3.1$ | $88.0 \pm 2.7$ | $89.3 \pm 3.3$ | $90.9 \pm 3.1$ | $76.6 \pm 2.0$ |
| | 20 | $78.9 \pm 4.4$ | $81.5 \pm 4.7$ | $83.3 \pm 4.7$ | $67.1 \pm 3.3$ | $88.1 \pm 2.7$ | $89.2 \pm 3.3$ | $90.9 \pm 3.1$ | $76.3 \pm 2.0$ |
| Deep | 5 | 84.5 | 87.2 | 86.8 | 85.0 | 91.3 | 92.5 | 92.2 | 91.5 |
| | 10 | 84.1 | 87.0 | 87.5 | 83.9 | 91.2 | 92.4 | 93.1 | 90.3 |
| | 20 | 83.7 | 86.6 | 87.2 | 84.1 | 91.0 | 92.2 | 92.9 | 90.6 |

**Table 12:** OOD detection results (TIM resize) trained on C100.

| Type | M | Conf. | TU | DU | KU | Conf. | TU | DU | KU |
|------|---|------|------|------|------|------|------|------|------|
| MC | 5 | $78.5 \pm 0.5$ | $80.6 \pm 0.3$ | $80.8 \pm 0.4$ | $76.6 \pm 0.6$ | $75.2 \pm 0.5$ | $78.1 \pm 0.6$ | $78.4 \pm 0.5$ | $70.9 \pm 1.1$ |
| | 10 | $78.7 \pm 0.6$ | $80.8 \pm 0.4$ | $81.0 \pm 0.5$ | $77.4 \pm 0.7$ | $75.4 \pm 0.6$ | $78.4 \pm 0.6$ | $78.7 \pm 0.5$ | $72.2 \pm 1.1$ |
| | 20 | $78.8 \pm 0.5$ | $80.9 \pm 0.4$ | $81.2 \pm 0.4$ | $77.9 \pm 0.7$ | $75.6 \pm 0.5$ | $78.4 \pm 0.4$ | $78.8 \pm 0.4$ | $72.9 \pm 1.4$ |
| Deep | 5 | 81.7 | 83.6 | 83.5 | 81.0 | 78.9 | 81.6 | 81.5 | 76.6 |
| | 10 | 82.3 | 84.1 | 84.2 | 82.4 | 79.8 | 82.2 | 82.4 | 78.7 |
| | 20 | 82.6 | 84.4 | 84.5 | 83.0 | 80.1 | 82.4 | 82.8 | 79.6 |

**Table 13:** OOD detection results (TIM random crop) trained on C100.

| Type | M | Conf. | TU | DU | KU | Conf. | TU | DU | KU |
|------|---|------|------|------|------|------|------|------|------|
| MC | 5 | $75.8 \pm 4.5$ | $78.8 \pm 4.8$ | $79.7 \pm 4.9$ | $69.3 \pm 3.7$ | $74.3 \pm 4.0$ | $78.5 \pm 4.5$ | $80.0 \pm 4.3$ | $60.8 \pm 3.7$ |
| | 10 | $75.7 \pm 4.8$ | $78.7 \pm 5.1$ | $79.7 \pm 5.2$ | $69.1 \pm 3.9$ | $74.2 \pm 4.2$ | $78.5 \pm 4.5$ | $80.2 \pm 4.5$ | $60.7 \pm 3.8$ |
| | 20 | $75.7 \pm 4.7$ | $78.6 \pm 5.0$ | $79.7 \pm 5.2$ | $69.0 \pm 4.1$ | $74.3 \pm 4.1$ | $78.4 \pm 4.4$ | $80.3 \pm 4.3$ | $60.6 \pm 4.4$ |
| Deep | 5 | 80.9 | 84.2 | 83.5 | 82.3 | 79.3 | 83.9 | 83.2 | 79.8 |
| | 10 | 82.8 | 86.5 | 85.7 | 85.5 | 81.0 | 85.8 | 85.0 | 83.7 |
| | 20 | 83.4 | 87.1 | 86.1 | 86.8 | 81.6 | 86.4 | 85.4 | 85.4 |

## C  BEHAVIOUR OF UNCERTAINTIES

This section investigates how the uncertainties produced from a vanilla Deep ensemble differ from self-distribution distilled derived systems, and how well hierarchical distribution distillation captures the behaviour of its teacher. The comparison will be made between the in-domain CIFAR-100 and, out of simplicity, only the out-of-domain SVHN test set.

Figure 4 shows the contrast of various uncertainties between an CIFAR-100 (ID) and SVHN (OOD) test sets. Clearly, the S2D systems output ID uncertainties in a consistent manner, even matching the

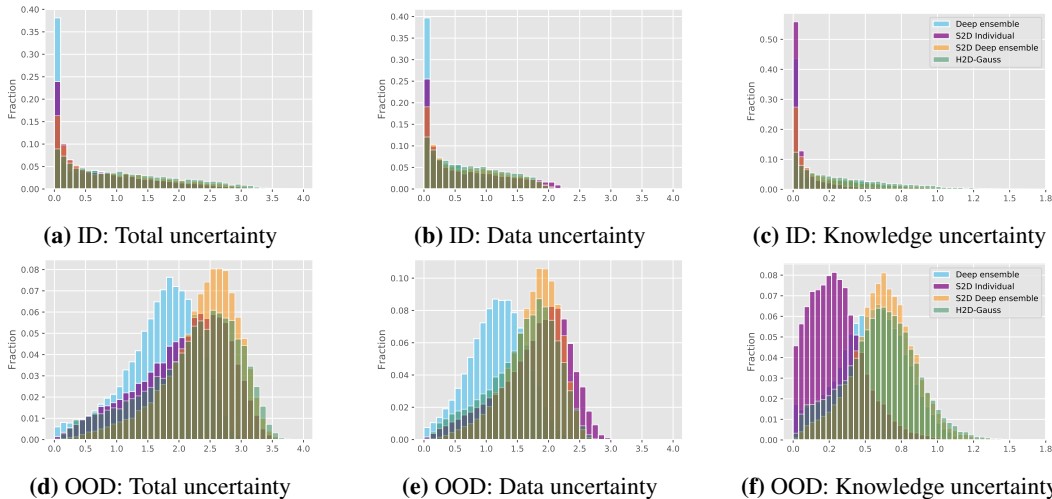

**Figure 4:** Histograms of various uncertainties produced by Deep ensemble, S2D, S2D Deep ensemble and H2D-Gauss systems. Out-of-distribution data was generated from the SVHN test set.

conceptually different Deep ensemble. Observe that S2D integrates temperature scaling (smoothing predictions) into the training of models; total and data uncertainties[3] estimated by these models will naturally have larger entropy than Deep ensembles. While it is expected that the Deep ensemble would have different behaviour on the SVHN OOD set, it is surprising to observe how well H2D-Gauss aligns with its S2D Deep ensemble teacher. An individual S2D model was also able to generate closely related total and data uncertainty estimates, but suffers significantly in producing consistent knowledge uncertainties. These results raise the question if a Gaussian student could capture the diversity in a vanilla Deep ensemble by modelling the logits, in a similar fashion to how H2D-Gauss models its teacher—a possible avenue for future work.

## D  ADDITIONAL EXPERIMENTS: WIDERESNET

Following the DenseNet-BC experiments in section 5 we repeated them with a different architecture. In this section we focus on a significantly larger WideResNet (Zagoruyko & Komodakis, 2016) model with a depth of 28 and a widening factor of 10. The standard and S2D models were both trained as described in Zagoruyko & Komodakis (2016), with the S2D specific parameters being the same as previously described. The only difference is that teacher predictions were generated using multiplicative Gaussian noise with a fixed standard deviation of $0.10$.

The H2D-Gauss model was also trained in a different manner. First, it was initialised from an S2D model trained for 150 epochs. Thereafter it was trained for an additional 80 epochs with a starting learning rate of $\eta = 2 \times 10^{-3}$ which was reduced by a factor of 4 after 60 epochs. For this section, EnD and H2D-Dir were not investigated.

Table 14 shows test set performance. Unlike previous experiments, S2D was not able to outperform an individual model by more than two standard deviations, in this case achieving around one standard deviation improvement in accuracy. Interestingly, the MC approach has worse accuracy for both the standard and S2D case, however this could be due to the small number of drawn samples ($M = 5$). Furthermore, both Deep ensembles significantly outperform their individual equivalents with the S2D version being slightly better in all measured performance metrics. The notable result in this table is the high performance of H2D-Gauss, able to outperform the Deep ensemble in C100 and achieve near ensemble performance in C100+.

In the OOD detection task we observe that both versions of the MC ensemble struggle to outperform their individual counterparts. There also seems to be a disparity in performance when comparing re-size and random cropped LSUN and TIM. With random crops, all S2D systems notably outperform

---

[3]Knowledge uncertainty does not necessarily increase with temperature.

**Table 14:** Test performance ($\pm$ 2 std).

| Dataset | C100 | | | C100+ | | |
|---|---|---|---|---|---|---|
| Model | Acc. | NLL | %ECE | Acc. | NLL | %ECE |
| Individual | 73.9 $\pm$ 0.5 | 1.05 $\pm$ 0.02 | 5.26 $\pm$ 0.78 | 81.1 $\pm$ 0.3 | 0.76 $\pm$ 0.01 | 5.21 $\pm$ 0.44 |
| S2D Individual | 74.2 $\pm$ 0.5 | 1.06 $\pm$ 0.05 | 5.48 $\pm$ 2.25 | 81.3 $\pm$ 0.3 | 0.74 $\pm$ 0.01 | 4.24 $\pm$ 0.74 |
| MC ensemble | 73.6 $\pm$ 0.5 | 1.05 $\pm$ 0.03 | 4.70 $\pm$ 0.88 | 81.0 $\pm$ 0.5 | 0.74 $\pm$ 0.01 | 3.29 $\pm$ 0.36 |
| S2D MC ensemble | 73.8 $\pm$ 0.4 | 1.03 $\pm$ 0.04 | 2.95 $\pm$ 1.01 | 81.0 $\pm$ 0.3 | 0.73 $\pm$ 0.01 | 1.99 $\pm$ 0.35 |
| Deep ensemble | 77.1 | 0.88 | 5.08 | 83.4 | 0.63 | 2.27 |
| S2D Deep ensemble | 77.9 | 0.86 | 4.52 | 83.6 | 0.63 | 1.84 |
| H2D-Gauss | 77.4 | 0.95 | 5.19 | 82.8 | 0.71 | 2.45 |

their standard counterparts. In this case both S2D Individual and H2D-Gauss were able to outperform the Deep ensemble using any uncertainty metric. In the other case of resizing LSUN and TIM images and in SVHN the detection performance difference is smaller but the S2D Deep ensemble still remains the best model with both H2D-Gauss and Deep ensemble performing similarly.

**Table 15:** LSUN (resize) OOD detection results. **Best** in column and **best** overall.

| Model | OOD %AUROC | | | | OOD %AUPR | | | |
|---|---|---|---|---|---|---|---|---|
| | Conf. | TU | DU | KU | Conf. | TU | DU | KU |
| Individual | 76.3 $\pm$ 0.5 | 76.7 $\pm$ 0.6 | | | 70.7 $\pm$ 0.8 | 71.1 $\pm$ 0.9 | | |
| S2D Individual | 76.0 $\pm$ 1.1 | 76.5 $\pm$ 1.5 | 76.7 $\pm$ 1.4 | 75.7 $\pm$ 1.6 | 71.4 $\pm$ 1.8 | 72.0 $\pm$ 2.7 | 72.8 $\pm$ 3.7 | 69.7 $\pm$ 2.0 |
| MC ensemble | 75.8 $\pm$ 0.6 | 76.2 $\pm$ 0.7 | 76.4 $\pm$ 0.7 | 65.2 $\pm$ 1.7 | 70.3 $\pm$ 1.0 | 70.5 $\pm$ 1.1 | 70.8 $\pm$ 1.2 | 56.2 $\pm$ 1.5 |
| S2D MC ensemble | 75.7 $\pm$ 1.0 | 76.4 $\pm$ 1.7 | 77.0 $\pm$ 1.6 | 75.2 $\pm$ 2.1 | 71.0 $\pm$ 1.6 | 71.6 $\pm$ 2.7 | 73.1 $\pm$ 3.8 | 69.6 $\pm$ 2.6 |
| Deep ensemble | 77.6 | 78.0 | 78.4 | 68.0 | 72.3 | 72.6 | 73.1 | 58.8 |
| S2D Deep ensemble | **77.7** | **78.5** | **79.3** | 76.8 | **73.2** | **74.1** | **75.9** | 71.3 |
| H2D-Gauss | 77.1 | 77.2 | 77.8 | **77.5** | 72.0 | 71.8 | 71.9 | **72.3** |

**Table 16:** LSUN (random crop) OOD detection results. **Best** in column and **best** overall.

| | | | | | | | | |
|---|---|---|---|---|---|---|---|---|
| Individual | 72.4 $\pm$ 5.0 | 73.9 $\pm$ 5.4 | | | 67.0 $\pm$ 2.9 | 68.7 $\pm$ 3.1 | | |
| S2D Individual | 75.8 $\pm$ 3.4 | 77.6 $\pm$ 4.3 | 77.9 $\pm$ 4.7 | **76.5** $\pm$ 4.6 | 70.5 $\pm$ 3.9 | 72.6 $\pm$ 4.9 | 74.4 $\pm$ 4.7 | **71.4** $\pm$ 5.5 |
| MC ensemble | 68.9 $\pm$ 5.6 | 70.3 $\pm$ 6.0 | 70.9 $\pm$ 6.2 | 50.8 $\pm$ 3.7 | 64.0 $\pm$ 3.0 | 65.2 $\pm$ 3.5 | 66.1 $\pm$ 3.6 | 45.7 $\pm$ 1.5 |
| S2D MC ensemble | 72.7 $\pm$ 3.2 | 74.5 $\pm$ 4.1 | 75.9 $\pm$ 4.3 | 72.0 $\pm$ 4.4 | 67.7 $\pm$ 3.3 | 69.7 $\pm$ 4.6 | 73.4 $\pm$ 4.4 | 65.7 $\pm$ 5.0 |
| Deep ensemble | 72.1 | 74.2 | 75.2 | 60.6 | 67.2 | 69.2 | 70.5 | 51.6 |
| S2D Deep ensemble | 75.5 | **78.4** | **80.0** | 75.4 | **70.7** | **73.9** | **77.2** | 69.0 |
| H2D-Gauss | **76.0** | 77.6 | 77.8 | 76.4 | 69.6 | 71.5 | 74.1 | 70.9 |

**Table 17:** SVHN OOD detection results. **Best** in column and **best** overall.

| | | | | | | | | |
|---|---|---|---|---|---|---|---|---|
| Individual | 80.1 $\pm$ 4.6 | 81.6 $\pm$ 4.4 | | | 88.3 $\pm$ 2.4 | 89.0 $\pm$ 2.3 | | |
| S2D Individual | 80.1 $\pm$ 4.4 | 81.6 $\pm$ 4.4 | 81.9 $\pm$ 4.8 | 81.4 $\pm$ 5.4 | 88.6 $\pm$ 2.3 | 89.2 $\pm$ 2.5 | 90.1 $\pm$ 2.5 | 87.8 $\pm$ 4.1 |
| MC ensemble | 77.6 $\pm$ 4.9 | 79.1 $\pm$ 4.5 | 79.7 $\pm$ 4.5 | 56.6 $\pm$ 2.5 | 86.9 $\pm$ 2.3 | 87.5 $\pm$ 2.2 | 88.0 $\pm$ 2.2 | 70.2 $\pm$ 1.2 |
| S2D MC ensemble | 77.3 $\pm$ 4.7 | 79.0 $\pm$ 4.8 | 80.1 $\pm$ 4.6 | 77.3 $\pm$ 5.6 | 87.1 $\pm$ 2.5 | 87.7 $\pm$ 2.7 | 89.6 $\pm$ 2.5 | 85.7 $\pm$ 3.9 |
| Deep ensemble | **81.5** | 83.4 | 84.0 | 68.3 | 89.2 | 89.9 | 90.4 | 77.9 |
| S2D Deep ensemble | 81.5 | **83.7** | **84.6** | **81.8** | **89.6** | **90.5** | **92.0** | **88.1** |
| H2D-Gauss | 81.5 | 82.1 | 83.2 | 80.6 | 88.6 | 88.4 | 90.5 | 87.1 |

**Table 18:** TIM (resize) OOD detection results. **Best** in column and **best** overall.

| | | | | | | | | |
|---|---|---|---|---|---|---|---|---|
| Individual | 79.7 ±0.4 | 80.5 ±0.4 | | | 75.9 ±0.5 | 76.9 ±0.5 | | |
| S2D Individual | 79.2 ±0.6 | 80.0 ±0.5 | 80.2 ±0.3 | 80.2 ±0.4 | 76.0 ±1.0 | 77.1 ±1.0 | 77.1 ±0.7 | 76.7 ±0.7 |
| MC ensemble | 79.8 ±0.4 | 80.6 ±0.3 | 80.7 ±0.4 | 68.3 ±1.7 | 76.1 ±0.7 | 77.0 ±0.6 | 77.1 ±0.6 | 59.5 ±1.6 |
| S2D MC ensemble | 79.4 ±0.6 | 80.3 ±0.7 | 80.2 ±1.0 | 80.1 ±0.7 | 75.9 ±0.9 | 77.1 ±1.0 | 77.2 ±1.1 | 76.8 ±0.6 |
| Deep ensemble | 81.8 | 82.7 | 82.7 | 72.5 | 78.4 | 79.3 | 79.2 | 64.1 |
| S2D Deep ensemble | **81.9** | **82.9** | **82.9** | **82.5** | **79.0** | **80.2** | **80.2** | **79.6** |
| H2D-Gauss | 80.9 | 81.4 | 81.4 | 81.5 | 77.4 | 79.0 | 78.9 | 78.0 |

**Table 19:** TIM (random crop) OOD detection results. **Best** in column and **best** overall.

| | | | | | | | | |
|---|---|---|---|---|---|---|---|---|
| Individual | 71.2 ±3.8 | 72.8 ±4.0 | | | 68.9 ±3.5 | 70.9 ±4.0 | | |
| S2D Individual | 73.1 ±3.0 | 74.9 ±3.6 | 76.3 ±3.9 | 75.9 ±3.4 | 71.4 ±1.7 | 73.7 ±2.2 | 74.5 ±2.4 | 73.4 ±2.4 |
| MC ensemble | 70.1 ±3.5 | 71.8 ±3.7 | 72.1 ±3.7 | 57.1 ±1.0 | 68.1 ±3.6 | 70.2 ±3.9 | 70.6 ±3.9 | 50.4 ±1.1 |
| S2D MC ensemble | 71.7 ±2.7 | 73.8 ±3.2 | 74.2 ±3.3 | 73.7 ±3.1 | 70.0 ±1.5 | 72.6 ±1.7 | 73.3 ±1.8 | 71.9 ±1.6 |
| Deep ensemble | 72.2 | 74.5 | 74.7 | 65.2 | 70.3 | 72.9 | 73.0 | 58.1 |
| S2D Deep ensemble | 74.3 | **77.0** | 77.3 | **77.1** | **72.6** | **75.9** | **76.2** | **75.5** |
| H2D-Gauss | **75.2** | 76.9 | **77.3** | 76.4 | 72.0 | 74.0 | 74.5 | 73.5 |

