# OpenReview forum: "Self-Distribution Distillation: Efficient Uncertainty Estimation"
_ICLR.cc/2022/Conference — ICLR 2022 Submitted_

### Official Review · Reviewer_Wumr · 2021-10-31

**Correctness:** 4
**Technical Novelty And Significance:** 3
**Empirical Novelty And Significance:** 3
**Recommendation:** 5
**Confidence:** 3

**Details Of Ethics Concerns:**

I think this paper does not have Ethics concerns.

**Main Review:**

Strengths:

This paper is clearly presented.

The proposed method is easy to be integrated into other methods and Experiments are done on different tasks to validate effectiveness.

weakness:

my only concern is the novelty of this method. S2D combines parameter sharing, stochastic regularization and distribution distillation.  These components exist in the literature. For example, parameter sharing is popular in online knowledge distillation methods like [1]; Also, in experiments, the authors did not compare with other distillation methods and did not show the model size and flops comparison.
[1] Online Knowledge Distillation via Collaborative Learning

**Summary Of The Paper:**

This paper contributes to neural network classifier training and uncertainty prediction.  It proposes a self-distribution distillation method that can train a single model to estimate uncertainties in an integrated training phase.  Also, it is flexible to be extended to build ensembles of models in the training phase and efficiently deployed in the test phase. Experiments are done on both classification and out-of-distribution detection tasks to show effectiveness.

**Summary Of The Review:**

This paper proposed an efficient uncertainty prediction. Experiments are done on two tasks to do evaluations. The paper is well written. However, since the proposed method is like a combination of existing techniques. The main concern is the novelty of the proposed method and whether it is fairly compared with SOTA.

---

> ### Author Response · Authors · 2021-11-19
> **Response to Reviewer Wumr**
>
> The novelty of our method lies in the fact that it does not require a separate ensemble to be trained. That it is able to estimate measures of uncertainty and diversity without a separate ensemble. We have also included additional baseline systems and shown how S2D can be combined with other methods to achieve even better performance. Computational cost of training/testing has also been added to the paper.

---

### Official Review · Reviewer_MF93 · 2021-11-01

**Correctness:** 2
**Technical Novelty And Significance:** 2
**Empirical Novelty And Significance:** 2
**Recommendation:** 5
**Confidence:** 3

**Main Review:**

Although this paper is a creative attack on a significant problem, I cannot recommend acceptance at this time.
My main reservations are to do with how the paper situates itself with respect to prior work and the theoretical representation of the ideas.

The prior work section could be improved.
1) Ensemble methods, variational inference, and SWAG are distinct methods. Ensemble methods (e.g., Lakshminarayanan et al. 2017) are as you describe. Monte Carlo dropout (MCDO) is a variational inference (VI) method, which makes a variational approximation. The predictive distribution of an MCDO variational approximation can be efficiently computed with a Monte Carlo expectation, which has some resemblances to an ensemble, but has a different principled justification. SWAG is an ammortized stochastic gradient Markov-chain Monte Carlo method, which is neither an ensemble nor a variational method.
2) You write that MCDO "generally does not perform as well" as deep ensembles, which should be supported by citation (and is a complicated claim which depends a lot on the resource constraints and problem definition).
3) Generally, the claim that different measures of uncertainty are grounded in different measures is best supported by some evaluation of where the noise comes from and what it might have to do with the prior which was chosen.
4) Minor points: Jordan et al. 1999 Introduction to Variational Inference is a better citation for variational inference; the Gal, 2016 thesis which you cite for uncertainty decomposition is less appropriate than Kendall and Gal 2017 "What Uncertainties Do We Need for Bayesian Deep Learning for Computer Vision"; the Amodei et al. 2016 paper you cite isn't really about fully autonomous driving and you'd be better off citing a paper that is in the opening paragraph

If I understand correctly, the main difference between your proposal and Ensemble Distribution Distillation is the choice to tie all parameters before the last layer?
You mention that this may have a beneficial regularizing effect.
Are you able to construct experiments that show that this is the source of the benefit?
Are there other elements of your contribution that might extend here also?

I would like to see more of a theoretical investigation of what kinds of uncertainty are actually represented by the estimates.
For example, it seems like the network is not able to express any uncertainty that corresponds to the fixed feature representation.

Reference results for DenseNet-BC are ~82% accuracy on CIFAR-10, which is significantly greater than any of the results you report.
This makes me wonder if there is a problem with tuning the baselines, which might make the rest of the results less reliable?
Do you know what might be going on there?

**Summary Of The Paper:**

The authors propose a way to construct and train neural networks which represent uncertainty which is efficient with respect to compute and memory at execution-time.
They propose learning a shared feature representation with an ensemble-head which is distilled into a single head during training, with the ensemble-head thrown away for deployment.

**Summary Of The Review:**

I am not able to recommend acceptance. I think the paper could use a more careful grounding in prior work which investigates the contributions and their theoretical motivations more carefully. I also have questions about the implementation of the baselines and whether there might be a reason the performance of baselines falls below those reported for the architecture.

---

> ### Author Response · Authors · 2021-11-15
> **Additional Clarification on Concerns**
>
> Many thanks for the comments on the paper. We plan to address the concerns raised by yourself and the other reviewers when we have completed the additional experimental contrasts mentioned. Prior to posting this more complete response it would be useful to get some additional clarification on your concerns about the baselines that we present in the paper. In particular your comment:
>
>  "Reference results for DenseNet-BC are ~82% accuracy on CIFAR-10, which is significantly greater than any of the results you report. This makes me wonder if there is a problem with tuning the baselines, which might make the rest of the results less reliable? Do you know what might be going on there?”
>
> The results we give in the paper are from CIFAR-100. From the DenseNet paper [1] (DenseNet-BC with 100 layers and k = 12):
>
> Without data augmentation: 75.85 and with data augmentation: 77.73
>
> The numbers we gave in the paper (mean and +/- 2std) areL
>
> Without data augmentation: 74.6 \pm 0.5 and with data augmentation: 77.5 \pm 0.2
>
>
> The numbers that we obtain seem reasonable, in particular as it is not clear exactly what number is being quoted as the single value in the DenseNet paper.
>
> If you could supply some additional information it will allow us to more fully address the concerns you have about our baselines
>
> [1] Gao Huang, Zhuang Liu, Laurens van der Maaten, and Kilian Q. Weinberger. Densely connected convolutional networks. In Conference on Computer Vision and Pattern Recognition, 2017.

---

> > ### Comment · Reviewer_MF93 · 2021-11-16
> > **Clarification**
> >
> > Hi - thanks for asking for more detail. I was looking at this PapersWithCode benchmarking
> >
> > https://paperswithcode.com/sota/image-classification-on-cifar-100
> >
> > which is on CIFAR-100 as you say, apologies for typo.

---

> > > ### Author Response · Authors · 2021-11-16
> > > **Clarification**
> > >
> > > Thank you for pointing us to the website. Having scanned through PapersWithCode, the number referenced for DenseNet-BC is 82.82% accuracy equaling 100 - 82.82 = 17.18% error rate. This performance number corresponds to the largest DenseNet-BC model analysed in [1] which has 190 layers and k = 40 resulting in a model with 25.6M parameters. Additionally this was achieved using data augmentation.
> > >
> > > Our results are based on a different drastically smaller DenseNet-BC model with 100 layers and k = 12 resulting in a model with 0.80M parameters, which is why our results are different to what has been referenced on the website. For reference the results corresponding to our chosen model can also be found in the 3rd last low in Table 2 in [1].
> > >
> > > [1] Gao Huang, Zhuang Liu, Laurens van der Maaten, and Kilian Q. Weinberger. Densely connected convolutional networks. In Conference on Computer Vision and Pattern Recognition, 2017.

---

> > > > ### Comment · Reviewer_MF93 · 2021-11-22
> > > > **Thanks**
> > > >
> > > > Thanks for clarifying - I appreciate it!

---

> ### Author Response · Authors · 2021-11-19
> **Response to Reviewer MF93**
>
> We would like to thank you for comments and hope the confusion between us regarding the performance of our baseline systems has been clarified, that the number reported on the PapersWithCode corresponds to a significantly larger DenseNet-BC model different to the one we chose to focus on in our paper.
>
> Regarding the regularising effect of S2D: The Table below shows how performance changes as you add the building blocks for self-distribution distillation.
>
> | Dataset                             	| C100     	| C100+    	|
> |-------------------------------------	|----------	|----------	|
> | Modle                               	| Accuracy 	| Accuracy 	|
> |                                     	|          	|          	|
> | Individual (Baseline)               	| 74.6     	| 77.5     	|
> |  + Gaussian Dropout                 	| 74.8     	| 77.8     	|
> |   + Self-Distillation               	| 75.5     	| 78.0     	|
> |    + Self-Distribution Distillation 	| 75.7     	| 78.1     	|
>
> Simply adding the stochastic regulariser (Gaussian Dropout) without performing self-distribution distillation improves performance marginally. Furthermore, if one simply self-distills the teacher predictions onto the student additional gain in performance is achieved. Self-distribution distillation then finally achieves the best performance.
>
> Regarding theoretical investigation into uncertainty: The standard performance metrics and discussion of uncertainty are currently included in the paper. A more theoretical analysis of the the kinds of uncertainty represented by the models is an interesting research question. However, the focus of the current paper is to present an efficient, in terms of training and inference, model that enables uncertainty measures to be obtained.

---

> > ### Comment · Reviewer_MF93 · 2021-11-29
> > **Thank you for your updates**
> >
> > I have increased the score I have assigned because you have satisfied most of my specific points (though I would still recommend further checking the discussion of prior work to make sure it is accurate). I think your updated version makes some significant strides relative to the earlier draft. I am currently highly borderline between 5/6, with the impression that the main thing stopping me from being more enthusiastic is a lack of significance and novelty. I would not object to acceptance if others were championing it, but neither will I advocate for it.
> >
> > I would recommend that for a resubmission you focus your exposition on the ways in which the contribution differs from a strong basline- given that ensembles and distillation are both widely used for their uncertainty properties, how can you analyse the *interaction* between them here and be confident that any performance improvements is due to the details of your method rather than the properties of those individual methods. That sort of analysis can be like the ablation you do here, but ideally also with some more theoretical grounding.

---

### Official Review · Reviewer_EMdF · 2021-11-02

**Correctness:** 3
**Technical Novelty And Significance:** 3
**Empirical Novelty And Significance:** 2
**Recommendation:** 5
**Confidence:** 5

**Main Review:**

*Strengths*

* It is a simple model that requires only one network and two "heads" (linear layers at the end of the network) for ensemble distillation.
* The method is able to match the deep ensemble's performance for Cifar-100 (classification, calibration), and LSUN and SVHN (OOD deteection).

*Weaknesses*

* Clarity: The notion could be improved and does not contribute to understanding the work. E.g., what's $M$, $m$, $\alpha$, $\delta$?,  There are several typos and missing words (e.g., "viewed from a Bayesian.", "over categoricals:", "There are a wide range of"). Please proofread the paper and fix these.
* Significance: The distillation itself is very similar to [1] and the authors could make a better job in detailing the difference of it. The difference is about using the self-distribution distillation, however, the authors fail to compare with existing methods to compare whether their methods outperform other distillation or ensembling methods.
* Insufficient evaluation: There are NO (!) comparisons to other distillation approaches or state-of-the-art (efficient) ensembling methods. Among others, works like [1, 2, 3, 4] can be compared to. Please look into these works for more models that can be compared to. Further, it is also good to not just compare the performance of the distilled model but also compare its efficiency (e.g., number of parameters, FLOPs).

*References*

[1] Malinin, A., Mlodozeniec, B. and Gales, M., 2019. Ensemble distribution distillation. arXiv preprint arXiv:1905.00076.

[2] Wenzel, F., Snoek, J., Tran, D. and Jenatton, R., 2020. Hyperparameter ensembles for robustness and uncertainty quantification. arXiv preprint arXiv:2006.13570.

[3] Havasi, M., Jenatton, R., Fort, S., Liu, J.Z., Snoek, J., Lakshminarayanan, B., Dai, A.M. and Tran, D., 2020. Training independent subnetworks for robust prediction. arXiv preprint arXiv:2010.06610.

**Summary Of The Paper:**

The paper proposes an ensemble distillation approach, in which one network is used as feature extractor and "two heads" (two networks representing teacher and student) are added to the network for self-distillation. The multiple teacher predictions can be generated through by adding multiplicative Gaussian noise. The distillation approach follows Malinin et al. [1] who proposes to model the predictive distribution with a Dirichlet and predicts the parameter of the Dirichlet distribution (instead of predicting the Categorial distribution). This approach can be both used for model distillation and ensemble distillation. The authors evaluated their models on CIFAR-100, LSUN, SVHN w.r.t. classification performance, calibration and out-of-distribution detection.

**Summary Of The Review:**

I do think this is an interesting approach of using just one model for ensembling and distillation. However, at this stage, I found the paper lacks in clarity, significance and thorough evaluation (see weaknesses above for details). Thus I am currently recommending a weak reject.

---

> ### Author Response · Authors · 2021-11-19
> **Response to Reviewer EMdF**
>
> As a clarification, we would like to point out that a general version of self-distribution distillation would require “two heads” but that our chosen configuration does not require any additional parameters to a standard model. We have also added extra clarifications of why S2D is an intriguing approach and its strengths. Additional comparisons to other methods have also been included. We would also like to point out that although the method has been termed “self-distribution distillation“ it does not require a separate ensemble to be trained and is a significantly faster to train compared standard distillation approaches. To compare distillation approaches we proposed “hierarchical distribution distillation” which can outperform a standard Deep Ensemble in many of the detection experiments.

---

> > ### Comment · Reviewer_EMdF · 2021-11-29
> > **Thank you for the rebuttal and updates.**
> >
> > Dear authors,
> >
> > thank you for updating the the paper, clarifying the novelty and adding additional comparisons. I do think the paper looks way better with these changes made. While I appreciate the additional results, I have some concerns that the comparison with MIMO (Havasi et al., 2021) is not fair. The table shows a very small MIMO model, and the original MIMO paper shows a much higher performance that would outperform this method. Further, I am also hesitant to increase my score due to the novelty and significance of this work. I would recommend a resubmission in which the authors can add fair comparisons and analyze their model more in detail what works and why. Nevertheless, I am not opposed against accepting this paper if my fellow reviewers would champion it.

---

> > > ### Author Response · Authors · 2021-11-30
> > > **Response to Reviewer EMdF**
> > >
> > > Dear reviewer,
> > >
> > > The strength of our approach is that it is not sensitive to model size, and unlike MIMO, does not require overly large models in order for it to work. Furthermore the results reported in the MIMO paper are based on significantly more expensive training approaches which according to [1] "makes the training cost of MIMO comparable to that of BatchEnsemble and Ensemble models". This makes the comparison to S2D unfair under their setting. However, both S2D and MIMO can be combined together and we show that this approach can produce even better uncertainties.
> > >
> > > [1] Marton Havasi, Rodolphe Jenatton, Stanislav Fort, Jeremiah Zhe Liu, Jasper Snoek, Balaji Lakshminarayanan, Andrew M. Dai, Dustin Tran. “Training independent subnetworks for robust prediction”. ICLR, 2021

---

### Official Review · Reviewer_MHeN · 2021-11-03

**Correctness:** 4
**Technical Novelty And Significance:** 3
**Empirical Novelty And Significance:** 3
**Recommendation:** 5
**Confidence:** 3

**Main Review:**

Strengths
- The paper is pretty solid in its technical idea contributions. It explores the (likely previously unexplored) space of modeling uncertainty with distillation/self-distillation very thoroughly, although using Dirichlet-based output for modeling uncertainty is not new.
- The writing is generally good. The good story flow in the paper makes it easy to read and follow.
- The proposed methods achieve good results compared to the baselines.

Weaknesses
- The paper mentions that ensembles suffer from high computational requirements and they propose S2D as an alternative to that. However, S2D Individual still has some pretty large performance gaps to Deep Ensemble, especially in Table 1 and 2. The methods that outperform Deep Ensemble still need heavy computations.
- The paper only includes EnD, traditional MC, and Deep Ensembles as the methods for experimental comparison. There are many more existing works that do uncertainty estimation. One example is `Learning for single-shot confidence calibration in deep neural networks through stochastic inferences. CVPR 2019.` that distills confidence/uncertainty knowledge to a student that does single-shot confidence estimation.
- En2D is introduced as an existing method that can distill the distribution knowledge to the student in Sec. 2.2 but it is not included in experimental comparison. Is EnD actually En2D?
- The tables do not show the computational (memory, inference time) requirements of each method/row. This makes it hard to make a good comparison between the methods.
- What are the differences between "Best in category" and "Best overall"?
- In the Introduction section, there is this statement "inputs it is expected that the trained model parameters can return reliable predictions." that does not make any sense.

**Summary Of The Paper:**

This paper introduces several distillation-based uncertainty estimation methods with distribution-based outputs and losses for the student network: (1) distilling from an ensemble of point-estimation/regular sub-network to a distribution-based Dirichlet sub-network (S2D); (2) distilling from an ensemble of Dirichlet networks to a single network (H2D) that either predicts a Dirichlet distribution (H2D-Dir) or a Gaussian distribution over multiple Dirichlet parameters (H2D-Gauss). Experimental results show that the proposed methods outperform conventional MC and deep ensemble methods on several tasks.

**Summary Of The Review:**

This paper introduces some nice ideas for uncertainty estimation with distillation, but it fails to make comparison to many more related existing works and it has some clarity issues. It is hard to know how it stands when it sufficiently considers the existing methods for comparison.

---

> ### Author Response · Authors · 2021-11-19
> **Response to Reviewer MHeN**
>
> Regarding point 1 (in Weaknesses) it is true that self-distribution distillation is not able to reach Deep ensemble performance but this is not a fair comparison due to a significant difference in computational costs at both train and test. However, even with this significant difference in resource requirements the S2D model can be quite competitive in certain detection tasks. For point (3) we have made it more clear why ensemble distribution distillation (En2D) [1] has not been included in the experiments. The issue with this approach is that it only works when the ensemble displays significant amount of diversity in its predictions and is not overconfident. Specifically, overconfidence can cause the gradients from the log-likelihood loss to explode [2] which is why it was not possible to train En2D systems using DenseNet-BC models. To avoid this issue the original paper [1] used a simple VGG-16 architecture which does not display overconfident predictions. Regarding point (5) what we meant by “category” was a comparison of only one of the uncertainty metrics {Conf, TU, DU, KU}. This has now been changed to “column” to make it clearer.
>
>
> [1] Andrey Malinin, Bruno Mlodozeniec, Mark Gales. “Ensemble Distribution Distillation”. ICLR, 2019.
>
>
> [2] Max Ryabinin, Andrey Malinin, Mark Gales. “Scaling Ensemble Distribution Distillation to Many Classes with Proxy Targets”. NeurIPS, 2021.

---

### Public Comment · ~Meet_P._Vadera1 · 2021-11-16
**Interesting work**

Hi,

This is an interesting paper! I enjoyed reading it :)

We have some past work around distilling Bayesian posterior distribution for uncertainty estimation as well as a benchmark paper looking at various uncertainty quantification ask (including OOD detection) that we think might be relevant to this paper [1, 2]- hope you find it useful!

1. Meet P. Vadera, Brian Jalaian, and Benjamin M. Marlin. Generalized bayesian
posterior expectation distillation for deep neural networks. In Proceedings of the 36th Conference on Uncertainty in Artificial
Intelligence (UAI), volume 124 of Proceedings of Machine Learning Research, pages
719–728. PMLR, 03–06 Aug 2020b. URL https://proceedings.mlr.press/v124/
vadera20a.htm

2. Meet P. Vadera, Adam D. Cobb, B Jalaian, and Benjamin M. Marlin. URSABench:
Comprehensive Benchmarking of Approximate Bayesian Inference Methods for Deep
Neural Networks. In ICML Workshop on Uncertainty and Robustness in Deep
Learning, 2020

---

### Author Response · Authors · 2021-11-19
**General Response**

We thank you for the time and effort in giving constructive reviews! We have taken into account your comments and made modifications to the paper. Most minor errors have been fixed, additional references have been added and certain clarifications have been made detailing why self-distribution distillation (S2D) is an attractive options and that it can easily be combined with many other methods in order to achieve even better performance and detection ability. A revised version has been uploaded which includes many more experiments such as comparisons to MIMO [1] and SWAG [2] and how well these methods perform when each one is combined with S2D. We also provide specific responses to each of you.


Below is an excerpt of additional results and computational costs associated with each approach:

| Dataset        	| C100     	| C100+    	|
|----------------	|----------	|----------	|
| Model          	| Accuracy 	| Accuracy 	|
|                	|          	|          	|
| Individual     	| 74.6     	| 77.5     	|
| MIMO (M = 2)   	| 75.2     	| 77.6     	|
| SWAG-Diag      	| 74.8     	| 77.7     	|
| MC ensemble    	| 75.6     	| -        	|
| S2D Individual 	| 75.7     	| 78.1     	|

| Model          	| Parameters 	| Inference 	|
|----------------	|------------	|-----------	|
|                	|            	|           	|
| Individual     	| 0.80M      	| 2.3ms     	|
| MIMO (M = 2)   	| 0.83M      	| 2.3ms     	|
| SWAG-Diag      	| 1.60M      	| 11.6      	|
| MC ensemble    	| 0.80M      	| 11.5      	|
| S2D Individual 	| 0.80M      	| 2.3       	|

All of the methods above have similar costs at training time, approximately 4.3h on a single NVIDIA V100. Inference time was computed per input.

Additionally each of these methods above are combined with S2D without any additional cost at inference. Results are shown in Tables 1-3 in the revised version.

[1] Marton Havasi, Rodolphe Jenatton, Stanislav Fort, Jeremiah Zhe Liu, Jasper Snoek, Balaji Lakshminarayanan, Andrew M. Dai, Dustin Tran. “Training independent subnetworks for robust prediction”. ICLR, 2021

[2] Wesley Maddox, Timur Garipov, Pavel Izmailov, Dmitry Vetrov, Andrew Gordon Wilson. “A Simple Baseline for Bayesian Uncertainty in Deep Learning”. NeurIPS 2019

---

### Decision · Program_Chairs · 2022-01-20

**Decision:**

Reject

**Comment:**

This article presents novel distillation-based methods for neural network training and uncertainty estimation. While the idea is interesting, there is a general agreement amongst reviewers that the paper lacks clarity, adequate discussion of the relevant literature  and comparisons to existing work. Although the revision uploaded by the authors goes in the right direction by adding some experiments and clarifying some of the issues raised by the reviewers, further work is needed to make the submission stronger.